# Epigenetic regulation of beta-endorphin synthesis in hypothalamic arcuate nucleus neurons modulates neuropathic pain in a rodent pain model

Yu Tao [1,7], Yuan Zhang[2,3,7], Xiaohong Jin[4,7], Nan Hua [1], Hong Liu[4], Renfei Qi [1], Zitong Huang [1], Yufang Sun [1,3], Dongsheng Jiang [5], Terrance P. Snutch[6], Xinghong Jiang[1,3] & Jin Tao [1,3] ✉

Although beta-endorphinergic neurons in the hypothalamic arcuate nucleus (ARC) synthesize beta-endorphin (β-EP) to alleviate nociceptive behaviors, the underlying regulatory mechanisms remain unknown. Here, we elucidated an epigenetic pathway driven by microRNA regulation of β-EP synthesis in ARC neurons to control neuropathic pain. In pain-injured rats miR-203a-3p was the most highly upregulated miRNA in the ARC. A similar increase was identified in the cerebrospinal fluid of trigeminal neuralgia patients. Mechanistically, we found histone deacetylase 9 was downregulated following nerve injury, which decreased deacetylation of histone H3 lysine-18, facilitating the binding of NR4A2 transcription factor to the miR-203a-3p gene promoter, thereby upregulating miR-203a-3p expression. Further, increased miR-203a-3p was found to maintain neuropathic pain by targeting proprotein convertase 1, an endopeptidase necessary for the cleavage of proopiomelanocortin, the precursor of β-EP. The identified mechanism may provide an avenue for the development of new therapeutic targets for neuropathic pain treatment.

Neuropathic pain is defined as pain caused by lesions and diseases of somatosensory nerves, as observed in the context of cancers, diabetes and traumatic nerve injury[1,2]. Therapeutic management of neuropathic pain has had limited success, as current medications such as non-steroidal anti-inflammatory drugs, gabapentinoids and opioids are modestly effective and/or produce severe adverse effects, including the potential for addiction and drug tolerance[3,4]. Identifying the cellular and molecular mechanisms driving chronic neuropathic pain, therefore, is of significant interest concerning the discovery of treatments

and preventative strategies[5,6]. The arcuate nucleus (ARC) of the hypothalamus, an evolutionarily conserved brain region located around the third ventricle in mammals, is a key component of the endogenous opioid peptide system[7]. Studies have shown that the ARC plays pivotal roles in the modulation of nociception[8,9] and damage to the ARC through chemical or electrical stimulation weakens the analgesic effect of morphine administration[10]. Importantly, β-endorphinergic neurons densely located in the ARC synthesize and release β-endorphin (β-EP) and are thought to participate in pain regulation[11]. For instance,

[1]Department of Physiology and Neurobiology & Centre for Ion Channelopathy, Suzhou Medical College of Soochow University, Suzhou 215123, PR China. [2]Department of Geriatrics & Clinical Research Center of Neurological Disease, The Second Affiliated Hospital of Soochow University, Suzhou 215004, PR China. [3]Jiangsu Key Laboratory of Neuropsychiatric Diseases, Soochow University, Suzhou 215123, PR China. [4]Department of Pain Medicine, The First Affiliated Hospital of Soochow University, Suzhou 215006, PR China. [5]Institute of Regenerative Biology and Medicine, Helmholtz Zentrum München, Munich 81377, Germany. [6]Michael Smith Laboratories and Djavad Mowafaghian Centre for Brain Health, University of British Columbia, Vancouver, BC V6T 1Z4, Canada. [7]These authors contributed equally: Yu Tao, Yuan Zhang, Xiaohong Jin. ✉e-mail: taoj@suda.edu.cn

patients with chronic neuropathic pain exhibited decreased levels of β-EP in their cerebrospinal fluid[12]. Moreover, the β-endorphinergic fiber bundles terminate in the midbrain periaqueductal gray matter[13], playing critical roles in the descending antinociceptive pathway[14]. Thus, understanding how pain-associated molecules in ARC neurons, particularly β-EP, are regulated after nerve injury may provide potential avenues for neuropathic pain management.

MicroRNAs (miRNAs) are a class of small single-stranded noncoding RNAs of approximately 22 nucleotides in length that can regulate gene expression at the posttranscriptional level by degrading their target mRNAs leading to translational repression[15]. An increasing number of studies have revealed that miRNAs participate in regulating almost every normal and many pathological cellular processes and that aberrant miRNA expression is a hallmark of multiple human disorders, including neuropathic pain[16]. Notably, we have previously demonstrated that replenishment of miR-32-5p in peripheral sensory neurons may serve as a potential therapeutic target to prevent the development of neuropathic pain[17]. It remains to be determined whether other miRNAs also play potential modulatory functions as well as any role of ARC β-endorphinergic neurons towards regulating nociceptive behaviors.

In the present study, we identify miR-203a-3p as a key functional noncoding RNA in ARC neurons that plays pivotal roles in the development and maintenance of nociceptive behavior. Downregulation of histone deacetylase 9 (HDAC9) decreased deacetylation of histone H3 lysine-18 (H3K18), facilitating the binding of NR4A2 transcription factor to the *miR-203a-3p* gene promoter and increasing miR-203a-3p expression levels following nerve injury. We also identify increased miR-203a-3p as underlying the maintenance of trigeminal neuropathic pain by targeting proprotein convertase 1 (PC1), an endopeptidase necessary for the cleavage of proopiomelanocortin (POMC) to the precursor of β-EP. Our findings highlight the critical role of epigenetic regulators and miR-203a-3p in regulating β-EP synthesis and their contribution to trigeminal-mediated neuropathic pain. This mechanistic understanding of endogenous opioid system may enable the discovery of therapeutic targets for the treatment of chronic neuropathic pain.

## Results

### miR-203a-3p is upregulated in ARC neurons after nerve injury
Trigeminal-mediated neuropathic pain was evoked by chronic constriction injury to the rat infraorbital nerve (CCI-ION). Animals exhibited a significant decrease ($p < 0.001$) in the mechanical threshold on Days 14, 21, and 28 after CCI-ION (Fig. 1a) compared to that following sham surgery. The expression of immediate early genes was analyzed from CCI-ION Day 14 to determine whether neuronal activity in ARC regions was altered. Notably, the number of neurons expressing Fos protein (Fos[+]) in the ARC was markedly elevated bilaterally following CCI-ION compared to the sham-operated groups (left, increased by 113.9 ± 29.5%; right, increased by 121.2 ± 22.3%; Fig. 1b). To identify microRNAs (miRNAs) and their regulatory networks associated with the pathogenesis of neuropathic pain, high-throughput transcriptome sequencing (RNA-seq) was performed on the rat ARC at Day 14 post-CCI-ION operation (accession number GSE216965). Among twenty-eight ARC-rich miRNAs exhibiting the highest increased expression (Fig. 1c), sixteen were identified as highly conserved across species, including human and rat, and thereafter, their expression in the ARC was confirmed by qPCR analysis (Fig. 1d). Four miRNAs, including miR-30d-5p, miR-199a-5p, miR-203a-3p and miR-451-5p, were upregulated by >100% in the bilateral ARC tissue of CCI-ION rats, with miR-203a-3p the most increased (313%, $p < 0.001$, Fig. 1d). We further examined the levels of miR-30d-5p, miR-199a-5p, miR-203a-3p and miR-451-5p in the cerebrospinal fluid (CSF) of patients with trigeminal neuralgia, and found that only miR-203a-3p was significantly increased when compared to that of age-matched healthy subjects (Fig. 1e). Subsequently,

we further explored the expression profile and biological function of miR-203a-3p in rats. qPCR analyses showed miR-203a-3p to be widely expressed throughout the rat brain, and although not statistically significant, miR-203a-3p in the ARC was comparatively higher relative to other brain areas associated with pain perception, such as the caudal trigeminal nucleus, periaqueductal gray matter and amygdala (Fig. 1f). In comparison to sham-operated groups, the expression levels of miR-203a-3p were markedly increased only in the ARC on Day 14 post-CCI-ION, but not in the nucleus accumbens (NAc) or hippocampus (HIP), although both areas possessed a high basal level of miR-203a-3p expression (Fig. 1g). In the peripheral sensory system, miR-203a-3p transcripts were detected in the trigeminal ganglia (TG) (Fig. 1h & Fig. S1), albeit at significantly lower levels than compared to miR-203a-3p expression in the rat ARC ( ~10.2-fold higher; Fig. 1i). The time course of miR-203a-3p expression in the ARC was also examined. In comparison to sham-operated groups, miR-203a-3p expression was markedly increased from Day 14 post-CCI-ION and maintained for at least 28 days in agreement with the escape threshold behavior time course (Fig. 1j). Fluorescence in situ hybridization (FISH) showed basal miR-203a-3p expression in the ARC, and that was increased bilaterally at Day 14 post-CCI-ION (Fig. 1k). FISH analysis combined with immunofluorescent staining revealed that miR-203a-3p-positive cells primarily colocalized with the neuronal marker NeuN, but rarely with the astrocyte marker glial fibrillary acidic protein (GFAP) or the microglial marker integrin CD11b (Fig. 1l). Statistical analysis showed that ~93.4% of the miR-203a-3p[+] neurons were labeled by NeuN, ~1.9% by GFAP, and ~3.7% by CD11b (Fig. 1l), indicating the dominant expression of miR-203a-3p in ARC neurons.

### miR-203a-3p regulates nociceptive behaviors
To determine whether miR-203a-3p directly participates in trigeminal-mediated neuropathic pain, we injected an antagomir of miR-203a-3p (antagomir-203a) or its negative control (antagomir-NC) bilaterally into the ARC on Day 14 post-CCI-ION. On Days 1-3 post drug administration antagomir-203a showed significant alleviation of mechanical allodynia ($p < 0.001$) while treatment with antagomir-NC did not elicit any improvement in escape threshold (Fig. 2a). Subsequently, we overexpressed miR-203a-3p target sequences by employing a neuron promoter-specific combinatorial lentiviral vector lenti-hSyn-miR-203a-3p-antisense (miR-203a-down) to investigate the effects of miR-203a-3p blockade on nociceptive behaviors. qPCR analysis revealed that no significant differences in the miR-203a-3p expression levels were observed in the ARC of miR-203a-down-treated CCI-ION rats (Fig. S2). Bilateral intra-ARC administration of miR-203a-down at Day 14 post-CCI-ION markedly alleviated mechanical allodynia from Day 3 through Day 10 post drug administration in both male (Fig. 2b) and female rats (Fig. 2c). We next examined the involvement of miR-203a-3p in the development of neuropathic pain by following the effect of miR-203a-down injection prior to CCI-ION surgery. Compared to that of sham-operated groups, pretreating rats with miR-203a-down significantly attenuated subsequent CCI-ION-induced mechanical allodynia at Day 14 post-surgery (Fig. 2d). We next explored whether mimicking the CCI-ION-induced upregulation of miR-203a-3p in ARC neurons of intact rats would affect nociceptive thresholds. As shown in Fig. 2e, bilateral administration of agomir-203a, but not agomir-NC, induced mechanical hypersensitivity and with a faster timeline compared to CCI-ION surgery. Directly increasing miR-203a expression in intact ARC neurons utilizing a lentiviral-mediated neuron-specific promoter lenti-hSyn-miR-203a-up (miR-203a-up) containing an eGFP-expression construct as a transfection marker showed eGFP fluorescence from Day 3 post-intra-ARC injection (Fig. 2f) and that was maintained for at least 21 days (Fig. S3). Notably, almost all GFP-expressing neurons with green fluorescence were located within the ARC (outlined by white dashed line), while the surrounding areas showed little expression, supporting the effects of drugs/reagents being specific to ARC (Fig. 2f).

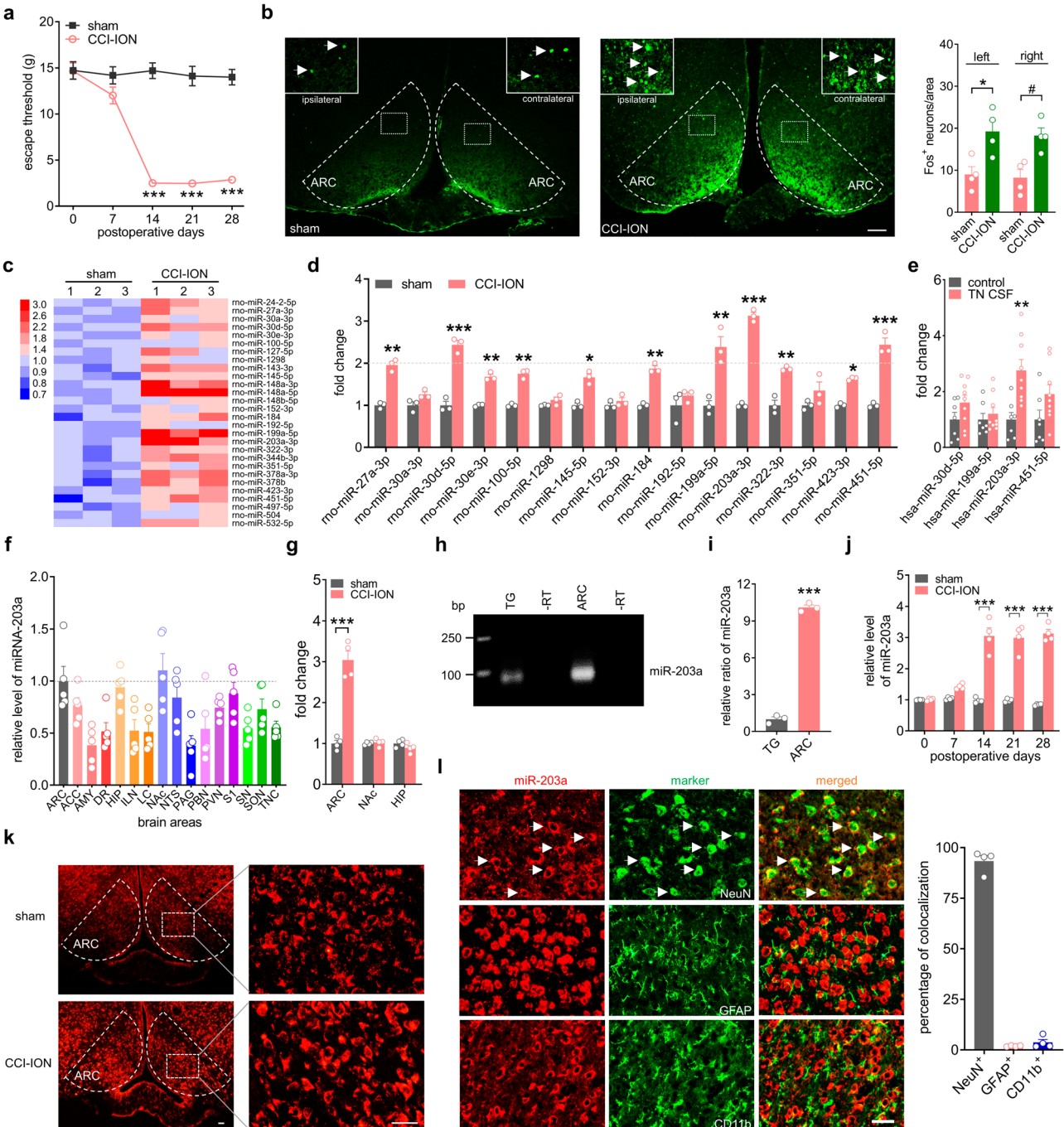

**Fig. 1 | miR-203a-3p is upregulated in ARC neurons after CCI-ION. a** Escape threshold in the sham- and CCI-ION-operated groups. ***$p < 0.001$ versus sham, two-way ANOVA followed by Bonferroni's test. $n = 10$ rats per group. **b** Protein expression of *c-fos* was analyzed in bilateral ARC regions (outlined by dashed line). Arrows show Fos+ ARC neurons. Bar charts show the quantification of Fos+ neurons. *$p = 0.0197$ versus sham, #$p = 0.0113$ versus sham, two-sided Student's *t* test. $n = 4$ rats per group. Scale bar, 100 μm. **c** Heatmap indicating the increased ARC-rich miRNAs in the ARC of CCI-ION rats. Color bar (*up left*) indicates the scale of standardized miRNA levels. $n = 6$ rats per group. **d** qPCR analysis of sixteen conserved miRNAs in the ARC. *$p < 0.05$, **$p < 0.01$, ***$p < 0.001$ versus sham, two-sided Student's *t* test. $n = 6$ rats per group. **e** Alterations in miR-30d-5p, miR-199a-5p, miR-203a-3p and miR-451-5p in the CSF of TN patients and aged-matched healthy subjects. **$p = 0.0031$ versus control, two-sided Student's *t* test. $n = 7$ subjects in control group and $n = 9$ subjects in TN group. **f** qPCR analysis of miR-203a-3p (miR-203a) expression in different brain areas, including the arcuate nucleus (ARC), anterior cingulate cortex (ACC), amygdala (AMY), dorsal raphe nucleus (DR), hippocampus (HIP), intralaminar nucleus (ILN), locus coeruleus (LC), nucleus

accumbens (NAc), nucleus of the solitary tract (NTS), periaqueductal gray (PAG), parabrachial nucleus (PBN), paraventricular nucleus (PVN), primary somatosensory cortex (S1), substantia nigra (SN), supraoptic nucleus (SON), and trigeminal nucleus caudalis (TNC). Data were normalized to miR-203a-3p expression in the ARC. $n = 5$ rats per group. **g** qPCR analysis of miR-203a level in the ARC, NAc, and HIP. ***$p = 0.0001$ versus sham, two-sided Student's *t* test. $n = 8$ rats per group. **h–i** RT–PCR (**h**) and qPCR analysis (**i**) of miR-203a-3p expression in the ARC and TG of intact rats. ***$p < 0.001$ versus TG group, two-sided Student's *t* test. $n = 6$ rats per group. **j** Time course of miR-203a-3p expression. ***$p < 0.001$ versus sham, one-way ANOVA followed by Bonferroni's test. $n = 8$ rats per time point per group. **k** Representative FISH images of miR-203a-3p expression in the bilateral ARC. $n = 3$ rats per group. Scale bar, 25 μm. **l** FISH analysis of miR-203a-3p (*red*) combined with immunostaining for NeuN, GFAP, or CD11b (*green*). Bar charts show the percentage of double-stained ARC neurons among total numbers of miR-203a-3p+-labeled neurons. Arrows show colocalization. $n = 4$ rats per group. Scale bar, 25 μm. All data are presented as mean values ± SEM.

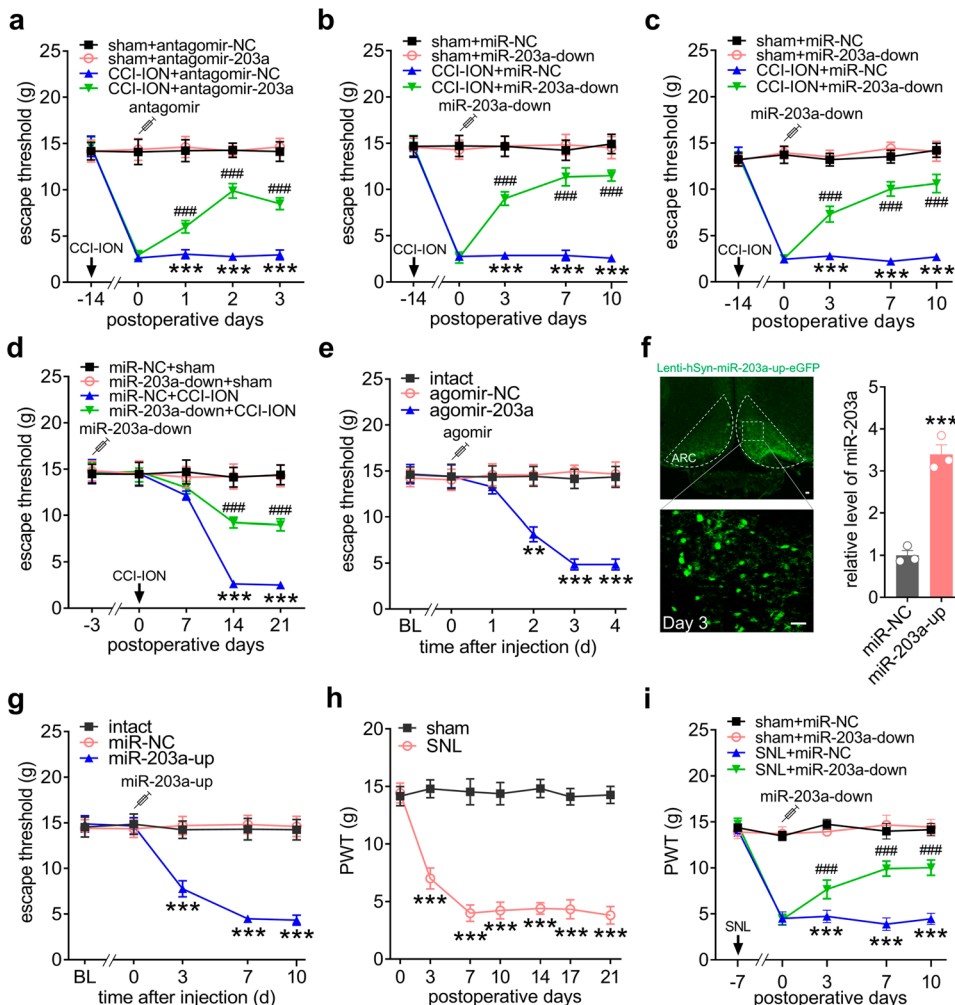

**Fig. 2 | miR-203a-3p regulates pain behaviors. a** Injection of antagomir-203a-3p (antagomir-203a) alleviated CCI-ION-induced mechanical allodynia. ***$p < 0.001$ versus sham + antagomir-NC, ###$p < 0.001$ versus CCI-ION + antagomir-NC, two-way ANOVA followed by Bonferroni's tests, $n = 8$ rats per group. **b, c** Administration of lenti-miR-203a-3p-down (miR-203a-down) alleviated CCI-ION-induced mechanical allodynia in male (**b**) or female rats (**c**). ***$p < 0.001$ versus sham + miR-NC, ###$p < 0.001$ versus CCI-ION + miR-NC, two-way ANOVA followed by Bonferroni's tests, $n = 8$ rats per group. **d** Pretreatment with miR-203a-down attenuated the CCI-ION-induced decrease in mechanical threshold. ***$p < 0.001$ versus miR-NC + sham, ###$p < 0.001$ versus miR-NC + CCI-ION, two-way ANOVA followed by Bonferroni's tests, $n = 10$ rats per group. **e** Administration of agomir-203a-3p (agomir-203a) in the ARC of intact rats induced mechanical hypersensitivity. **$p < 0.01$, ***$p < 0.001$ versus agomir-NC, two-way ANOVA followed by Bonferroni's tests, $n = 8$ rats per

group. **f** *Left*, fluorescence image of eGFP-expressing ARC neurons 3 days after injection of lenti-hSyn-miR-203a-3p-up (miR-203a-up). Scale bar, 25 μm. *Right*, qPCR analysis of miR-203a-3p on day 3 after miR-203a-up administration. ***$p = 0.0007$ versus miR-NC, two-sided Student's $t$ test. $n = 6$ rats. **g** Administration of miR-203a-up in intact rats induced mechanical hypersensitivity. ***$p < 0.001$ versus miR-NC, two-way ANOVA followed by Bonferroni's tests. $n = 10$ rats per group. **h** Paw withdrawal threshold (PWT) in the sham- and SNL-operated groups. ***$p < 0.001$ versus sham, two-way ANOVA followed by Bonferroni's test. $n = 8$ rats per group. **i** Administration of miR-203a-down alleviated SNL-induced mechanical allodynia. ***$p < 0.001$ versus sham + miR-NC, ###$p < 0.001$ versus SNL + miR-NC, two-way ANOVA followed by Bonferroni's tests, $n = 8$ rats per group. All data are presented as mean values ± SEM.

Similarly, qPCR analysis showed that the expression of miR-203a-3p was significantly increased on Day 3 post-injection for miR-203a-up but not for miR-NC (Fig. 2f). Moreover, bilateral intra-ARC administration of miR-203a-up into intact rats induced significant mechanical hypersensitivity from Day 3 that was maintained for more than 10 days, while similar treatment with miR-NC did not affect pain behavior (Fig. 2g). Further, we examined whether the regulation of miR-203a-3p on CCI-ION-induced mechanical allodynia would be extended to other different neuropathic pain models including spinal nerve ligation (SNL) models. As shown in Fig. 2h, rats exhibited a significant decrease ($p < 0.001$) in the paw withdrawal threshold (PWT) from Day 3 through Day 21 after SNL compared to the counterparts following sham surgery. Bilateral intra-ARC injection of miR-203a-down at Day 7 post-SNL significantly attenuated mechanical allodynia from Day 3 through Day 10 post drug administration (Fig. 2i). Together, these behavioral data

demonstrate that miR-203a-3p plays pivotal roles in the modulation of neuropathic pain.

## NR4A2 is a transcriptional activator of miR-203a-3p

Initiation of gene transcription is governed by transcription factors (TFs)[18]. Thus, to elucidate TFs that regulate miR-203a-3p expression, we performed a promoter deletion analysis of the approximately 2000 bp 5′-region upstream of the *miR-203a-3p* gene (bp −1978 to bp 0). A series of pGL3 luciferase reporter plasmids containing fragments of the *miR-203a-3p* promoter region were compared with that of the intact promoter (pGL3-F1 to F5; Fig. 3a). Examining relative luciferase activity showed that only the pGL3-F5 construct (bp -1978 to bp -1275) exhibited a significant decrease compared with pGL3-F4 (Fig. 3b). These results suggested that the upstream fragment from bp -1605 to bp -1275 (termed ΔF) might contain *cis*-regulatory elements critical for

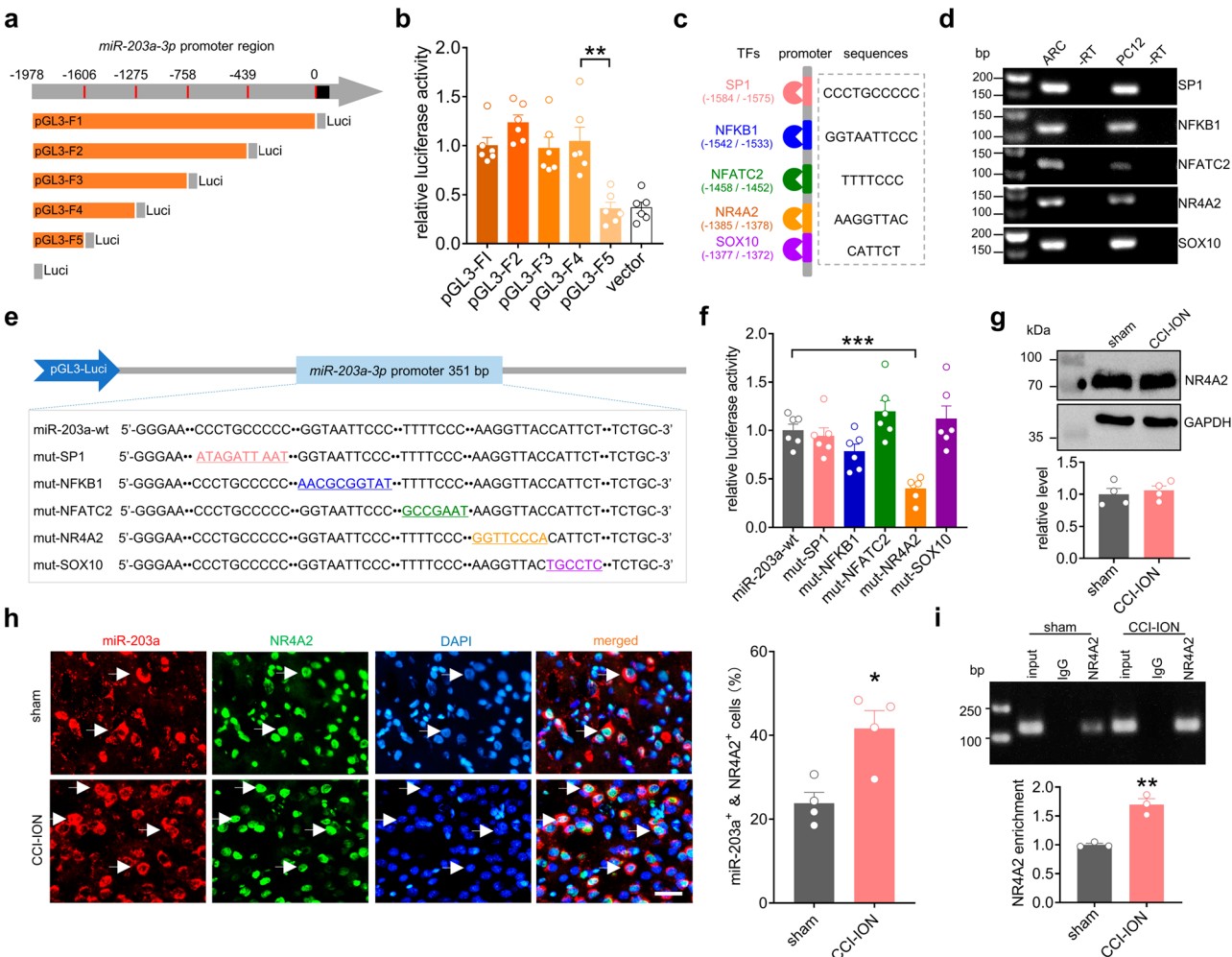

**Fig. 3 | NR4A2 is a transcriptional activator of miR-203a-3p. a** Schematic diagrams of distinct constructs of pGL3 luciferase reporters (pGL3-F1 to F5). **b** Luciferase reporter assay of the transcriptional activities of pGL3-F1 to F5 as indicated in (**a**). **p = 0.0015 versus pGL3-F4, two-sided Student's *t* test. Data are from six independent experiments. **c** The JASPAR algorithm predicted that ΔF in the *miR-203a-3p* promoter region harbors five putative protein binding elements, including SP1, NFKB1, NFATC2, NR4A2, and SOX10. **d** RT–PCR analysis of above five transcription factors in the ARC of intact rats as well as in PC12 cells. Samples without reverse transcriptase (-RT) were used as negative controls. Similar results were obtained in four independent experiments and the data shown were from one representative experiment. **e** Schematic diagrams of the luciferase reporter constructs including mut-SP1, mut-NFKB1, mut-NFATC2, mut-NR4A2, and mut-SOX10. **f** Luciferase activities of miR-203a-wt, mut-SP1, mut-NFKB1, mut-NFATC2, mut-

NR4A2 and mut-SOX10. ***p = 0.0003 versus miR-203a-wt, two-sided Student's *t*-test. Data are from six independent experiments. **g** Immunoblot analysis of the protein abundance of NR4A2 in the ARC. n = 8 rats per group. **h** FISH analysis of miR-203a-3p (*red*) combined with immunostaining for NR4A2 protein (*green*) and DAPI (*blue*) in the rat ARC on Day 14 following sham or CCI-ION operation. Bar charts show the percentage of both miR-203a⁺ and NR4A2⁺ ARC neurons among total numbers of DAPI-labeled cells. *p = 0.0116 versus sham, two-sided Student's *t*-test. n = 4 rats per group. Arrows show colocalization. Scale bar, 25 μm. **i** ChIP–qPCR analysis indicating the binding of NR4A2 to the *miR-203a-3p* gene promoter on Day 14 following sham or CCI-ION operation. The fold enrichment of ChIP data was normalized to the input DNA and then compared to the sham group. Data are from three independent experiments. **p = 0.0025 versus sham, two-sided Student's *t*-test. n = 6 rats per group. All data are presented as mean values ± SEM.

*miR-203a-3p* gene expression. Employing the computational algorithm JASPAR (http://jaspar.genereg.net) to predict TFs possessing binding sites within the ΔF region of the *miR-203a-3p* promoter we identified five potential factors: specificity protein 1 (SP-1), nuclear factor kappa B subunit 1 (NFκB1), nuclear factor of activated T cells 2 (NFATC2), nuclear receptor subfamily 4 group A member 2 (NR4A2), and SRY-box transcription factor 10 (SOX10) (Fig. 3c). RT–PCR data revealed that all five TFs were endogenously expressed in the rat ARC and also in rat PC12 cells (Fig. 3d & Fig. S4). To test for functional interactions between the TFs and the ΔF region of the *miR-203a-3p* gene promoter, a series of luciferase reporter gene constructs containing mutations within the relevant TF binding sites were generated: mut-SP-1, mut-NFKB1, mut-NFATC2, mut-NR4A2 and mut-SOX10 (Fig. 3e). Of these, only the mut-NR4A2 resulted in a significant reduction (p < 0.001) in luciferase activity (Fig. 3f). Further analysis employing the UCSC

Genome database showed that the NR4A2 binding-site residues are conserved across human, rat and mouse. To examine whether NR4A2 levels correlated with the increased expression of miR-203a-3p immunoblot analyses were performed. Surprisingly, immunoblot analysis of rat ARC lysates at Day 14 post-CCI-ION revealed no significant difference in protein expression of NR4A2 compared to that after sham surgery (Fig. 3g & Fig. S5). FISH analysis combined with immunofluorescence labeling confirmed that NR4A2 was heavily colocalized with miR-203a-3p in rat ARC neurons at Day 14 in both sham and CCI-ION animals (Fig. 3h). While miR-203a-3p expression was increased in the rat ARC on Day 14 following CCI-ION (increased by 78.4 ± 17.3%), immunofluorescent staining revealed that the expression of NR4A2 appeared unchanged (increased by −5.8 ± 4.5%) (Fig. 3h). Statistical analysis showed that 23.8 ± 2.6% of DAPI-labeled ARC cells were positive for both miR-203a-3p and NR4A2 in sham

surgery, while 41.7 ± 4.3% in CCI-ION rats (Fig. 3h). Given the above, we hypothesized that the CCI-ION driven change miR-203a-3p expression might result from alteration of NR4A2 binding to the *miR-203a-3p* gene promoter. Chromatin immunoprecipitation-PCR (ChIP-PCR) analysis using lysates from rat ARC showed a significant enrichment of the *miR-203a-3p* promoter from anti-NR4A2 complexes at Day 14 post-CCI-ION compared to sham groups (Fig. 3i & Fig. S6) that were not immuno-precipitated when IgG was used as a negative control. Overall, at Day 14 post-CCI-ION the promoter region of the *miR-203a-3p* gene exhibits a higher enrichment of NR4A2 occupancy compared to that after sham surgery (Fig. 3i).

### HDAC9 downregulation facilitates the binding of NR4A2 to the miR-203a-3p promoter

DNA methylation and histone modifications are two main epigenetic markers controlling gene expression through interfering with tran-scription factor binding[19]. Notably, bioinformatic analysis with Meth-Primer (www.urogene.org) did not identify any CpG islands within the ΔF region of the *miR-203a-3p* gene promoter (Fig. S7), ruling out the possibility of CpG DNA methylation involved in any possible epige-netic regulation. Histone lysine methylation and acetylation are the two major posttranslational modifications of histone proteins in chromatin architecture and play pivotal roles in the modulation of miRNA gene transcription[20]. In particular, acetylation of histone H4 (H4ac) or of histone H3 lysine, including H3K9ac, H3K14ac, H3K18ac, and H3K27ac, correlates with transcriptional activation[21]. As such, we examined the expression of these acetylated histone isoforms post-CCI-ION finding that ARC H3K18ac protein levels were significantly upregulated at Day 14 after CCI-ION compared with that after sham surgery (Fig. 4a & Fig. S8), while comparatively H4ac and other acetylated isoforms of histone H3, including H3K9ac, H3K14ac, and H3K27ac remained unchanged (Fig. 4a). Subsequent FISH analysis combined with immunofluorescence labeling demonstrated that H3K18ac was located almost exclusively in the nucleus of ARC neurons (stained by DAPI), and was coexpressed with miR-203a-3p in both sham- and CCI-ION-operated rats (Fig. 4b). Statistical analysis showed that 22.6 ± 2.8% of DAPI-labeled ARC cells were positive for both miR-203a-3p and H3K18ac in sham surgery, while 42.3 ± 4.1% in CCI-ION rats (Fig. 4b). Notably, most ARC neurons expressing high levels of H3K18ac also exhibited strong miR-203a-3p expression in CCI-ION rats (Fig. 4b). Moreover, ChIP-PCR analysis from ARC lysates showed increased amplification of a *miR-203a-3p* promoter fragment from anti-H3K18ac complexes compared to IgG control (Fig. 4c & Fig. S9). On Day 14 post-CCI-ION, the promoter region of the *miR-203a-3p* gene exhibited significantly higher enrichment of H3K18ac occupancy compared to sham (~2.2-fold increase; $p < 0.001$; Fig. 4c). H3K18 is known to be acetylated by histone acetylases including EP300, CBP, KAT2A and KAT6A, and deacetylated by histone deacetylases, includ-ing HDAC9, SIRT2 and SIRT7[22]. Examining the expression of these enzymes in Day 14 sham and post-CCI-ION using qRT-PCR showed that HDAC9 mRNA expression was significantly decreased following CCI-ION (Fig. 4d) while the other histone acetylases/deacetylases showed no significant changes (Fig. 4d). At the protein level, HDAC9 expres-sion was also found to be downregulated in the ARC of CCI-ION ani-mals (Fig. 4e & Fig. S10). Moreover, examination of ARC sections by immunostaining showed that HDAC9 was strongly colocalized with NeuN (~91.2%) and comparatively less colocalized with GFAP (~2.2%) or CD11b (~5.8%) (Fig. 4f).

Next, we determined whether the observed HDAC9-mediated H3K18ac regulation participated functionally in nerve injury-induced mechanical allodynia. Local induction of HDAC9 in ARC neurons using a lentiviral-mediated neuron-specific promoter lenti-hSyn-HDAC9-up construct (HDAC9-up) abrogated the CCI-ION-induced upregulation of both H3K18ac protein (Fig. 4g & Fig. S11) and miR-203a-3p expres-sion (Fig. 4h). Further, bilateral ARC administration of HDAC9-up of

CCI-ION rats decreased the binding of NR4A2 to the *miR-203a-3p* gene promoter region (Fig. 4i & Fig. S12). Notably, HDAC9-up treatment also alleviated Day 14 post-CCI-ION induced mechanical allodynia (Fig. 4j). A chemically modified siRNA-mediated knockdown of HDAC9 (HDAC9-siRNA) was further employed to examine the role of HDAC9 in nociceptive behaviors. Intra-ARC application of HDAC9-siRNA in intact rats significantly increased H3K18ac protein levels (Fig. 4k & Fig. S13), upregulated miR-203a-3p expression (Fig. 4l) and enhanced the binding of NR4A2 to the *miR-203a-3p* gene promoter (Fig. 4m & Fig. S14). Finally, bilateral intra-ARC administration of intact rats with HDAC9-siRNA, but not its negative control (NC-siRNA), produced significant mechanical hypersensitivity from Day 2 through Day 8 after injection (Fig. 4n).

### miR-203a-3p targets the endopeptidase PCSK1

miRNAs generally regulate gene expression posttranscriptionally through binding to the 3′-UTR of their target mRNAs[15]. To identify miR-203a-3p target genes correlated with neuropathic pain, we predicted potential candidate genes by the combined use of TargetScan (http://www.targetscan.org), miRanda (http://www.microrna.org/), miRDB (www.mirdb.org/) and miRwalk (http://www.ma.uni-heidelberg.de/apps/zmf/mirwalk/) resulting in 13 candidates by the intersection of the four prediction algorithms (Fig. 5a). Among these candidates, qPCR analysis indicated that the expression levels of six genes were more than 50% downregulated in CCI-ION rats with *PCSK1* being the most robustly downregulated (Fig. 5b). Interestingly, the PC1 endo-peptidase encoded by *PCSK1* may contribute to nociceptive mechan-isms by cleavage of POMC to β-lipotropin (β-LPH), the precursor of endogenous opioid peptide β-EP in ARC neurons[23]. To determine whether PCSK1 is directly targeted by miR-203a-3p, luciferase repor-ters were constructed by inserting the 3′-UTR of PCSK1 containing the wild-type (PCSK1-wt) or the mutant of miR-203a-3p targeting sequence (PCSK1-mut) downstream of the firefly luciferase gene (Fig. 5c). Com-pared to that of cells transfected with negative control, the PCSK1-wt reporter showed significantly decreased luciferase activity when cotransfected with the miR-203a-3p mimics (Fig. 5d), and this inhibi-tion was rescued by the mutant reporter PCSK1-mut (Fig. 5c, d). The miR-203a-3p-binding sites predicted by using TargetScan (http://www.targetscan.org) are highly conserved across vertebrate (Fig. 5e), suggesting their functional importance in the regulation of PCSK1 by miR-203a-3p. We next determined whether miR-203a-3p regulation affected the functional expression of PCSK1 in ARC neurons. FISH analysis combined with immunofluorescent labeling showed that miR-203a-3p colocalized with PC1 in ARC neurons in both sham- and CCI-operated rats (Fig. 5f, g). Most ARC neurons expressing high levels of miR-203a-3p appeared to exhibit a modest expression level of PC1 at Day 14 of post CCI-ION rats (Fig. 5f, g). Both intra-ARC administration of agomir-203a (Fig. 5h & Fig. S15) and local induction of miR-203a expression by miR-203a-up (Fig. 5i & Fig. S16) induced a significant decrease in PC1 protein expression in the ARC of intact rats. Together, these findings demonstrated that miR-203a-3p directly targets the 3′-UTR of *PCSK1* mRNA to regulate PC1 expression.

### PC1 underlies miR-203a-3p-mediated nociceptive behaviors

We next determined whether PC1 participates in the regulation of neuropathic pain. Compared with that after sham surgery, the abun-dance of ARC PC1 protein was significantly decreased from Day 14 through Day 28 after CCI-ION, while levels of the PC2, another endo-peptidase cleaving β-LPH to β-EP, did not exhibit any significant change (Fig. 6a & Fig. S17). Sham surgery also did not result in changes in the abundance of PC1 or PC2 proteins (Fig. 6b & Fig. S17). The decreased mRNA level of PCSK1 at day 0, 7, 14, 21, and 28 (Fig. S18) showed an inverse correlation with miR-203a-3p upregulation in the ARC of CCI-ION rats (Fig. 6c). Moreover, PC1 was highly colocalized with the neuronal marker NeuN (~94.9%) but rarely with GFAP (~1.1%)

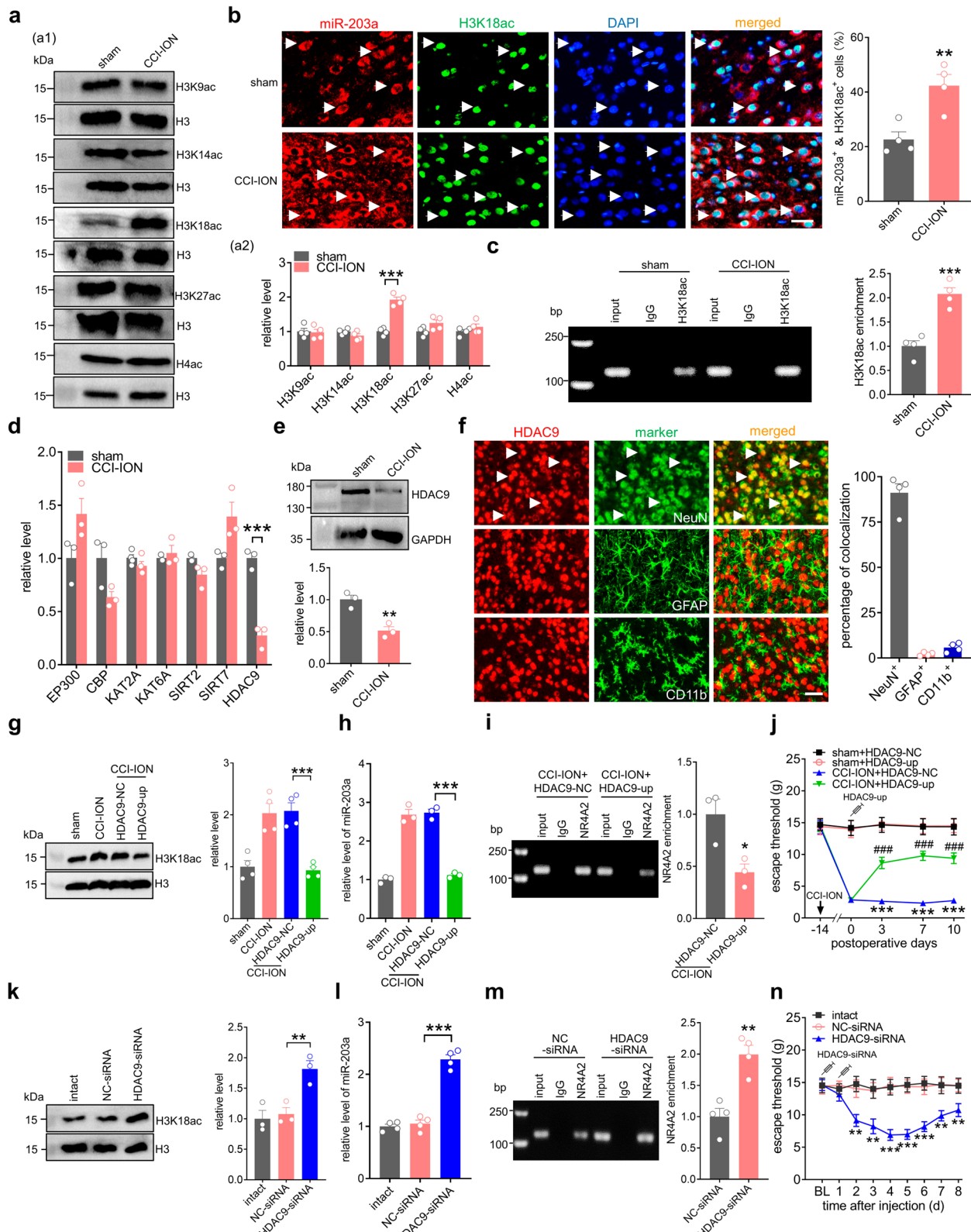

or CD11b (~3.5%) (Fig. 6d). Intra-ARC injection of PCSK1-up at Day 14 post-CCI-ION significantly increased expression of PC1 protein (Fig. 6e & Fig. S19). Further assessing whether PC1 is involved in neuropathic pain behaviors revealed that PCSK1-up microinjection attenuated nerve injury-induced mechanical allodynia from Day 3 through Day 10 after PCSK1-up administration in both male (Fig. 6f) and female rats (Fig. 6g). We next examined the involvement of PC1 towards the

maintenance of chronic neuropathic pain by evaluating the regulatory effect of PCSK1-up preinjection on CCI-ION-mediated mechanical allodynia. Compared to that of sham-operated rats, pretreating animals with PCSK1-up considerably alleviated CCI-ION-induced mechanical allodynia at Day 14 after surgery (Fig. 6h). Moreover, PCSK1-up intra-ARC treatment alleviated SNL-induced mechanical allodynia from Day 3 through Day 10 after PCSK1-up administration

**Fig. 4 | HDAC9 downregulation promotes miR-203a-3p expression.**
**a** Representative immunoblots (*a1*) and bar charts (*a2*) indicating the protein abundance of H3K9ac, H3K14ac, H3K18ac, H3K27ac, and H4ac. ***$p$ = 0.0001 versus sham, two-sided Student's *t*-test. $n$ = 8 rats per group. **b** FISH analysis of miR-203a-3p (*red*) combined with immunostaining for H3K18ac (*green*) and DAPI (*blue*). Bar charts show the percentage of both miR-203a⁺ and H3K18ac⁺ ARC neurons among total numbers of DAPI-labeled cells. Arrows show colocalization. **$p$ = 0.0076 versus sham, two-sided Student's *t*-test. $n$ = 4 rats per group. Scale bar, 25 μm. **c** ChIP–qPCR analysis indicating the binding of H3K18ac to the *miR-203a-3p* gene promoter. Data are from four independent experiments. ***$p$ = 0.0009 versus sham, two-sided Student's *t*-test. $n$ = 8 rats per group. **d** qPCR analysis of EP300, CBP, KAT2A, KAT6A, SIRT2, SIRT7, and HDAC9. ***$p$ = 0.0009 versus sham, two-sided Student's *t*-test. $n$ = 6 rats per group. **e** Immunoblot analysis of HDAC9. **$p$ = 0.0073 versus sham, two-sided Student's *t*-test. $n$ = 6 rats per group. **f** Double immunostaining of HDAC9 (*red*) with NeuN, GFAP, or CD11b (*green*) in ARC sections of intact rats. Bar charts show the percentage of double-stained ARC neurons among total numbers of HDAC9⁺-labeled neurons. Arrows show colocalization. $n$ = 4 rats per group. Scale bar, 25 μm. **g–i** Effects of lenti-hSyn-HDAC9-up (HDAC9-up) on H3K18ac protein expression (**g**, $n$ = 4 rats per group, $p$ = 0.0004), miR-203a-3p levels (**h**, $n$ = 6 rats per group, $p$ = 0.0001), and the binding of NR4A2 to the *miR-203a-3p* promoter (**i**, $n$ = 6 rats per group, $p$ = 0.0285) in the ARC on Day 14 after CCI-ION. ***$p$ < 0.001 versus CCI-ION + HDAC9-NC, one-way ANOVA followed by Bonferroni's tests. **j** Bilateral injection of HDAC9-up alleviated mechanical allodynia in rats 14 days after CCI-ION. ***$p$ < 0.001 versus sham + HDAC9-NC, ###$p$ < 0.001 versus CCI-ION + HDAC9-NC, two-way ANOVA followed by Bonferroni's tests. $n$ = 8 rats per group. **k–m** Effects of HDAC9 siRNA (HDAC9-siRNA) on H3K18ac protein expression (**k**, $n$ = 6 rats per group, $p$ = 0.0062), miR-203a-3p level (**l**, $n$ = 8 rats per group, $p$ = 0.0001), and the binding of NR4A2 to the *miR-203a-3p* promoter (**m**, $n$ = 8 rats per group, $p$ = 0.0024) in the ARC of intact rats. **$p$ < 0.01, ***$p$ < 0.001 versus NC-siRNA, one-way ANOVA followed by Bonferroni's tests. **n** Bilateral injection of HDAC9-siRNA induced mechanical hypersensitivity in intact rats. **$p$ < 0.01, ***$p$ < 0.001 versus NC-siRNA, two-way ANOVA followed by Bonferroni's tests. $n$ = 8 rats per group. All data are presented as mean values ± SEM.

(Fig. 6i). Exploring whether mimicking the CCI-ION-induced downregulation of PC1 in ARC neurons of intact rats alters the nociceptive thresholds, bilateral intra-ARC injection of chemically modified PCSK1-siRNA was performed. PCSK1-siRNA alone both decreased PC1 protein levels (Fig. 6j & Fig. S20) and produced marked mechanical hypersensitivity (Fig. 6k). We next examined whether PC1 expression was affected by manipulating miR-203a-3p or HDAC9 levels. Inhibition of miR-203a-3p by intra-ARC injection of miR-203a-down prevented the CCI-ION-induced decrease in PC1 protein expression (Fig. 6l & Fig. S21). Similar results were obtained with local induction of HDAC9 in ARC neurons by HDAC9-up administration (Fig. 6m & Fig. S22). To validate whether PC1 actually mediates miR-203a-3p-induced nociceptive behaviors, we locally administered PCSK1-up simultaneously with miR-203a-down in ARC neurons of CCI-ION rats. PCSK1-up was administered intra-ARC on Day 0 then miR-203a-down was administered on Day 7. Compared with the negative control groups (PCSK1-NC), treatment with PCSK1-up significantly alleviated CCI-ION-induced mechanical allodynia (Fig. 6n). Further administration of miR-203a-down (Day 7) did not elicit any additive effect on mechanical threshold in CCI-ION rats treated with PCSK1-up, as evidenced by the stability of the mechanical threshold from Day 7 through Day 14 after intra-ARC injection of miR-203a-down (Fig. 6n). In contrast, in CCI-ION rats treated with PCSK1-NC, administration of miR-203a-down induced a significant increase in the mechanical threshold ($p$ < 0.001; Fig. 6n). In summary, miR-203a-3p, by targeting PCSK1, regulates neuropathic pain behaviors in the CCI-ION model.

## Decreased β-EP contributes to mechanical allodynia following CCI-ION

PC1 cleaves POMC to β-LPH, which is further cleaved to β-EP by PC2 in ARC neurons[24] (Fig. 7a). We determined whether POMC expression was changed in trigeminal-mediated neuropathic pain. qPCR analysis showed that the mRNA level of POMC remained unchanged in rat ARC on Day 14 after CCI-ION (Fig. S23). However, our results showed that POMC protein expression was significantly increased in the rat ARC from 7 days to 28 days post-CCI-ION, but remained unaffected in the sham-operated groups (Fig. 7b, c & Fig. S24). Further ELISA analysis of ARC tissues showed that β-LPH (Fig. 7d) levels as well as its cleavage product β-EP (Fig. 7e) were significantly decreased in CCI-ION rats. ARC immunostaining indicated that PC1 was almost exclusively expressed in β-EP-positive ARC neurons in sham- and CCI-ION-operated rats (Fig. 7f). Notably, β-EP-negative ARC cells did not express PC1 (stained by nuclear marker DAPI). Statistical analysis showed that approximately 25.6 ± 1.9% of DAPI-labeled ARC cells were positive for both β-EP and PC1 in sham-operated rats, while 14.1 ± 1.5% in CCI-ION rats (Fig. 7f). Next, we examined whether manipulation of miR-203a-3p, PCSK1 and HDAC9 affected β-EP in the ARC of CCI-ION rats. Intra-ARC

administration of PCSK1-up (Fig. 7g), HDAC9-up (Fig. 7h), or antagomir-203a (Fig. 7i) 14 days after CCI-ION prevented the CCI-induced decrease in β-EP levels. Moreover, we explored whether ARC β-EP expression was effected by mimicking nerve injury-induced PCSK1 downregulation, HDAC9 downregulation, or increased miR-203a-3p. Intra-ARC injection of chemically modified PCSK1-siRNA (Fig. 7j), HDAC9-siRNA (Fig. 7k), or agomir-203a (Fig. 7l) all significantly decreased β-EP levels. We also examined whether β-EP mediates the miR-203a-3p-induced nociceptive response. Since β-EP is the cleavage product of POMC, we applied siRNA knockdown of POMC to abolish β-EP synthesis. The chemically modified POMC-siRNA was administered intra-ARC on Days 0, 3, and 6, and miR-203a-down was administered on Day 3. Administration of POMC-siRNA (Day 0) produced mechanical hypersensitivity on Day 3 in sham-operated rats (Fig. 7m). Compared to the sham-operated counterparts, CCI-ION rats treated with either POMC-siRNA or NC-siRNA (Day 0) exhibited significantly decreased mechanical thresholds (Fig. 7m). Further intra-ARC injection of miR-203a-down (Day 3) markedly attenuated mechanical allodynia in CCI-ION rats pretreated with NC-siRNA, while CCI-ION rats subjected to POMC-siRNA elicited no such effect (Fig. 7m), demonstrating that miR-203a-3p participates in neuropathic pain behaviors by regulating β-EP synthesis in ARC neurons. In support of this hypothesis, we examined whether the global blockade of μ-opioid receptors would prevent miR-203a-3p-mediated nociceptive responses. Intra-ARC injection of miR-203a-down at Day 14 days post-CCI-ION significantly alleviated mechanical allodynia (Fig. 7n). Subsequent intravenous administration of the specific μ-opioid receptor antagonist CTAP (Day 7) completely abolished the miR-203a-down-mediated alleviation of mechanical hyperalgesia in CCI-ION-operated rats. This effect lasted for at least 3 hours and recovered 6 hours after CTAP application (Fig. 7n). Together, these results demonstrate that miR-203a-3p-mediated regulation of β-EP synthesis could be targeted for the treatment of neuropathic pain after peripheral nerve injury.

## Discussion
In this study, we identified epigenetic mechanisms underlying β-EP synthesis in ARC neurons that contribute to neuropathic pain behaviors, and revealed that the expression of miR-203a-3p was increased both in the ARC of CCI-ION rats and in the CSF of TN patients. Inhibition of miR-203a-3p (pharmacologically or genetically) in the ARC effectively attenuated nerve injury-induced mechanical allodynia. Mechanistically, HDAC9 downregulation induced H3K18ac enrichment to facilitate the binding of NR4A2 to the promoter region of the *miR-203a-3p* gene and increased miR-203a-3p expression after nerve injury. Moreover, miR-203a-3p targets PCSK1, in turn affecting β-EP synthesis through cleavage of POMC (see Fig. 8). Thus, miR-203a-3p-mediated regulation of PC1 in ARC neurons plays pivotal roles in the endogenous

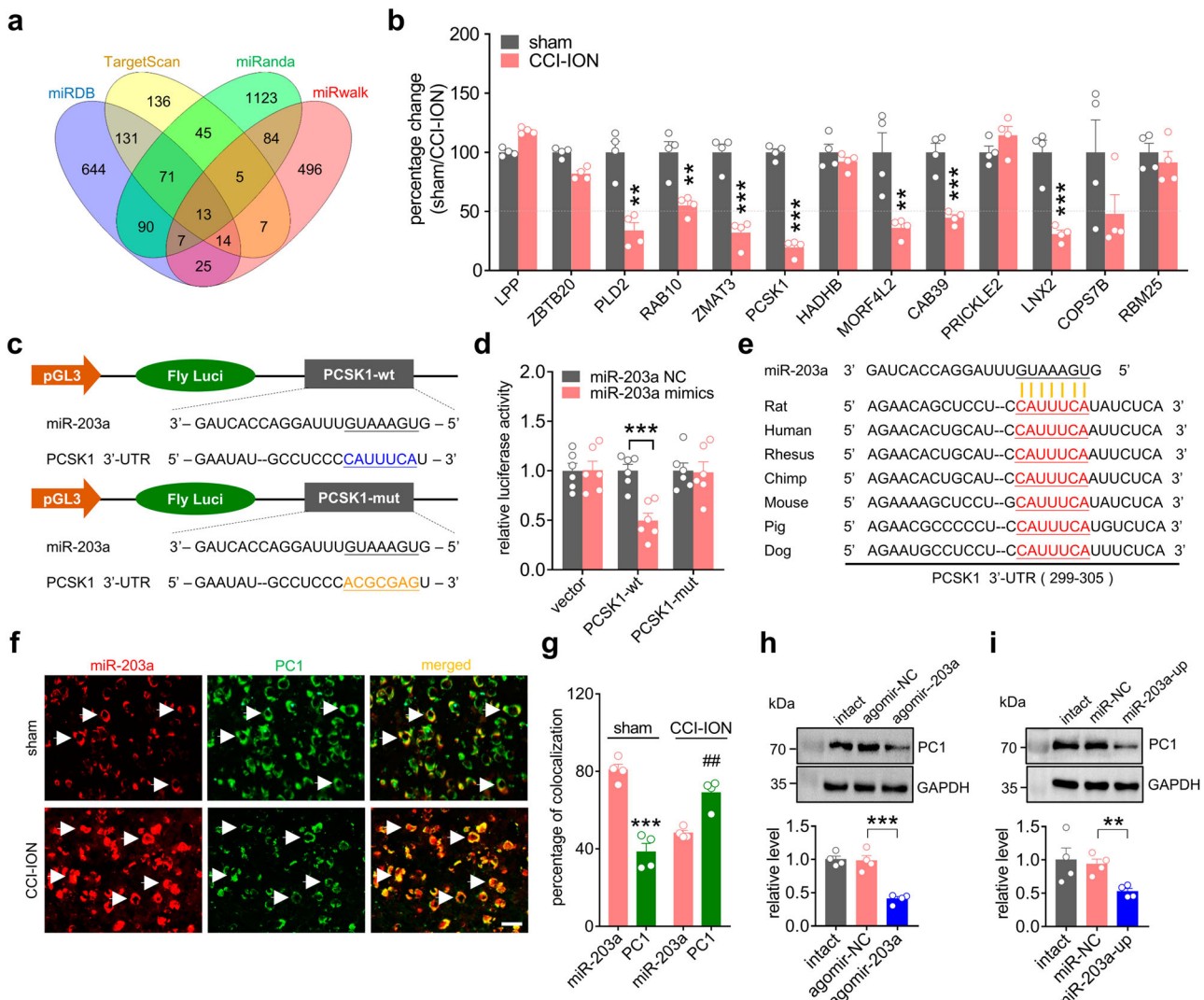

**Fig. 5 | miR-203a-3p targets *PCSK1*. a** Venn diagrams showing thirteen putative target genes of miR-203a-3p in the intersection predicted by the combined use of the "TargetScan" (*yellow*), miRDB (*blue*), "miRanda" (*green*) and "miRwalk" (*pink*) algorithms. **b** qPCR analysis indicating that six of the predicted target genes are downregulated by more than 50% in the ARC of rats at Day 14 post-CCI-ION-operation. \*\*$p < 0.01$, \*\*\*$p < 0.001$ versus sham, two-sided Student's *t*-test. $n = 8$ rats for each group. **c** Diagrams illustrate luciferase reporters containing wild-type (PCSK1-wt) or mutant PCSK1 (PCSK1-mut). **d** The luciferase reporter assay shows that miR-203a-3p interacts with the 3'-UTR of PCSK1. \*\*\*$p = 0.0005$ versus miR-203a NC, one-way ANOVA followed by Bonferroni's test. Data are from six independent experiments. **e** Diagrams illustrate sequences of PCSK1 3'-UTRs with highly conserved miR-203a-3p-binding sites in various vertebrates. **f** FISH analysis of the DIG-labeled miR-203a-3p probe (*red*) combined with immunofluorescent labeling with PC1 (*green*) in ARC sections 14 days after CCI-ION or sham surgery. Arrows show colocalization. $n = 4$ rats per group. Scale bar = 25 μm. **g** Bar charts show the percentage of double-stained ARC neurons among total numbers of miR-203a-3p+- or PC1+-labeled neurons. \*\*\*$p = 0.0002$ versus miR-203a + sham, ##$p = 0.0019$ versus miR-203a + CCI-ION, two-sided Student's *t* test. $n = 4$ rats per group.

**h, i** Representative immunoblots and bar charts showing that intra-ARC injection of agomir-203a (**h**) or miR-203a-up (**i**) decreased PC1 expression in the ARC of intact rats. \*\*$p = 0.0039$ versus miR-NC, \*\*\*$p = 0.0002$ versus agomir-NC, one-way ANOVA followed by Bonferroni's test. $n = 4$ rats per group. All data are presented as mean values ± SEM.

opioid system concerning the pathogenesis of neuropathic pain. Manipulation of miR-203a-3p as well as its relevant epigenetic factors may offer therapeutic targets for the treatment of neuropathic pain.

Human imaging studies revealed that the NAc plays pivotal roles in both acute and chronic pain states[25]. Indeed, inactivating the NAc with lidocaine diminishes tactile allodynia in the spared nerve injury model of neuropathic pain[26]. In addition, manipulation of the HIP participates in both the processing and the modification of nociception[27,28]. However, it should be noted that β-EP-producing neurons are mainly clustered in the ARC[29], while comparatively fewer in the HIP[30] or NAc[31]. Importantly, although NAc and HIP had a high basal level of miR-203a-3p expression, comparative to ARC, the expression levels of miR-203a-3p remained unchanged in NAc and HIP

regions after CCI-ION treatment. As a result, although both brain areas have potential involvement in pain regulation, the miR-203a-3p-mediated β-EP regulation seems less likely to function importantly in both nuclei to regulate neuropathic pain. MiRNAs regulate gene expression posttranscriptionally by binding the 3'-UTRs of their target mRNAs[15]. Increasing evidence supports tissue-specific changes in miRNA expression under distinct pain conditions such as persistent inflammatory, neuropathic, and cancer nociceptive behaviors[32,33]. However, little is known about how miRNAs themselves are modulated during pathophysiological processes. Here, we found that peripheral nerve injury significantly increased the H3K18ac levels at the miR-203a-3p promoter and that these modifications participated in regulating neuropathic pain. The acetylated histone H3K18, but not H4ac or other

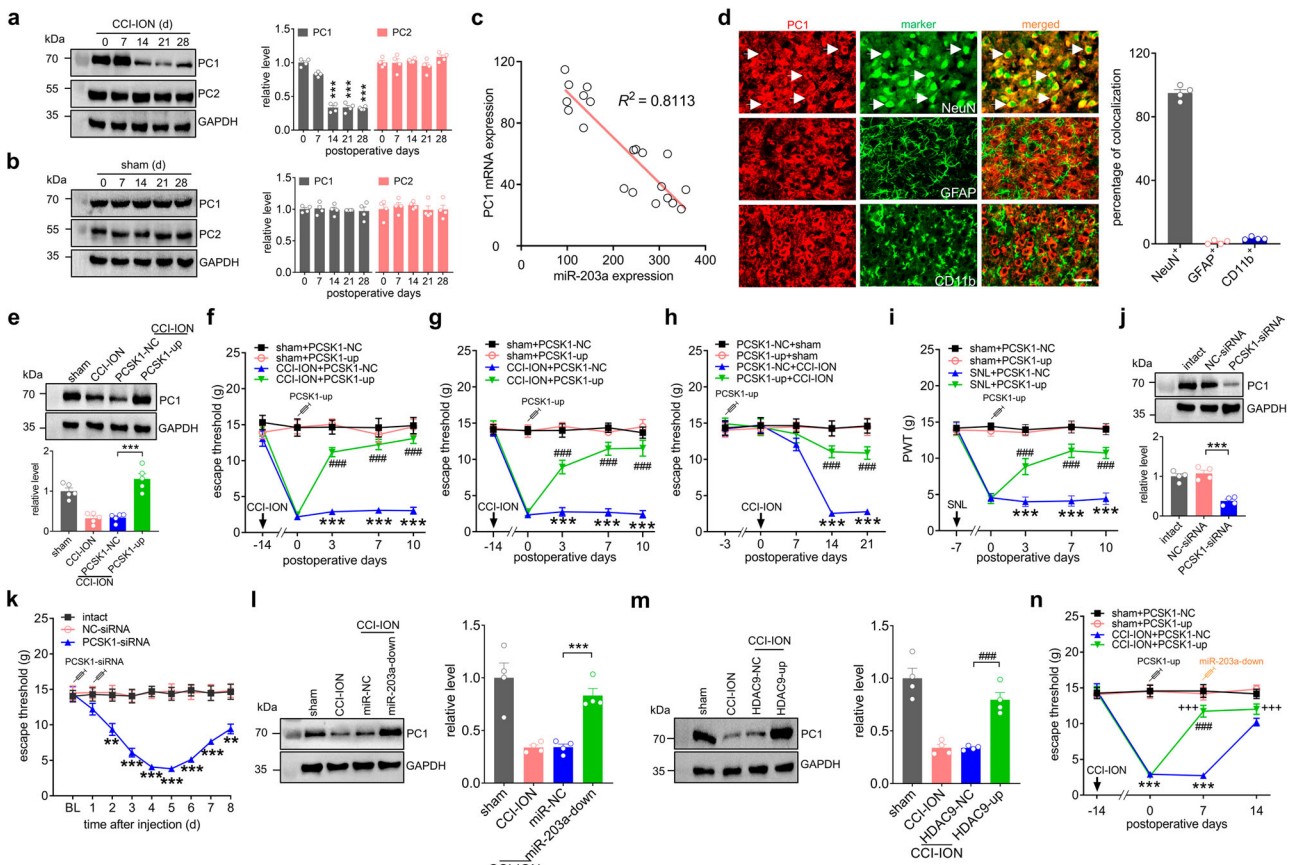

**Fig. 6 | PC1 is responsible for miR-203a-3p-mediated nociceptive behaviors.**
**a**, **b** Time course of PC1 and PC2 protein expression in the rat ARC after CCI-ION (**a**) or sham surgery (**b**). ***$p < 0.001$ versus PC1 on Day 0, one-way ANOVA followed by Bonferroni's test. $n = 8$ rats per time point per group. **c** Correlation between miRNA and PCSK1 mRNA expression ($R^2 = 0.8113$). **d** Immunostaining analysis of PC1 (*red*) with NeuN, GFAP, and CD11b (*green*) in ARC sections of intact rats. Bar charts show the percentage of double-stained ARC neurons among total numbers of PC1⁺-labeled neurons. Arrows show colocalization. $n = 4$ rats per group. Scale bar, 25 µm. **e** Administration of lenti-hSyn-PCSK1-up (PCSK1-up) restored the CCI-ION-induced decrease in PC1 expression. ***$p = 0.0001$ versus CCI-ION + PCSK1-NC, two-sided Student's $t$ test. $n = 5$ rats per group. **f**, **g** Injection of PCSK1-up alleviated CCI-ION-induced mechanical allodynia in male (**f**) or female (**g**) rats. ***$p < 0.001$ versus sham + PCSK1-NC, ###$p < 0.001$ versus CCI-ION + PCSK1-NC, two-way ANOVA followed by Bonferroni's test. $n = 8$ rats per group. **h** PCSK1-up pretreatment abolished the CCI-ION-induced decrease in mechanical threshold. ***$p < 0.001$ versus PCSK1-NC + sham, ###$p < 0.001$ versus PCSK1-NC + CCI-ION, two-way ANOVA followed by

Bonferroni's test. $n = 8$ rats per time point per group. **i** Bilateral administration of PCSK1-up alleviated SNL-induced mechanical allodynia. ***$p < 0.001$ versus sham + PCSK1-NC, ###$p < 0.001$ versus SNL + PCSK1-NC, two-way ANOVA followed by Bonferroni's test. $n = 8$ rats per group. **j** Administration of PCSK1 siRNA (PCSK1-siRNA) reduced PC1 expression in the ARC of intact rats. ***$p = 0.0003$ versus NC-siRNA, one-way ANOVA followed by Bonferroni's test. $n = 4$ rats per group. **k** Application of PCSK1-siRNA in intact rats induced mechanical hypersensitivity. **$p < 0.01$, ***$p < 0.001$ versus NC-siRNA, two-way ANOVA followed by Bonferroni's test. $n = 10$ rats per group. **l**, **m** Effects of miR-203a-down (**l**) or HDAC9-up (**m**) on the decreased PC1 expression induced by CCI-ION. ***$p = 0.0006$ versus CCI-ION + miR-NC, ###$p = 0.0006$ versus CCI-ION + HDAC9-NC, one-way ANOVA followed by Bonferroni's test. $n = 4$ rats per group. **n** Evaluation of preinjection of PCSK1-up on the miR-203a-down-induced attenuation of mechanical allodynia in CCI-ION rats.
***$p < 0.01$ versus sham + PCSK1-NC, ###$p < 0.001$ versus CCI-ION + PCSK1 on Day 7, ⁺⁺⁺$p < 0.001$ versus CCI-ION + PCSK1-up on Day 0, two-way ANOVA followed by Bonferroni's test. $n = 9$ rats per group. All data are presented as mean values ± SEM.

---

histone H3 acetylation, in the miR-203a-3p promoter was responsible for promoting NR4A2 binding to *miR-203a-3p* and mediating an increase in miR-203a-3p expression. Consistent with this, it has been demonstrated that trigeminal nerve root compression enhanced the ratio of immunoreactive H3K9ac-, H3K18-, and H3K27ac-positive cells in the rat trigeminal root entry zone[34], while another study reported that H3K9ac levels in the trigeminal ganglia were lower using the same animal model[35]. In addition, in an animal model of subchronic stressor-induced visceral hypersensitivity, the expression of H3K18ac and H3K9ac proteins, but not H3K27ac, was significantly increased in the dorsal spinal cord of rats and shown to play pivotal roles in pain regulation[36]. Interestingly, mice with chronic postsurgical pain were associated only with increased spinal cord levels of H3K27ac[37]. Although further investigation is required, these discrepancies may be attributed to the fact that histone modifications likely vary in different tissue/cell types expressing distinct acetylases and deacetylases[38], and further that different neuropathic pain models may exacerbate this

variability. For example, the increased H3K18ac enrichment mediated by the histone acetyltransferase CBP/p300 generally causes transcriptional activation of nucleotide synthesis enzyme genes[39], while decreased H3K18ac occurs through the histone deacetylase Sirt7[40].

Chromatin remodeling regulates gene expression by increasing or decreasing histone acetylation via histone acetyltransferase and deacetylase (HDAC) activities[41]. However, the role of HDAC9-driven deacetylase activity in the context of brain diseases is not well understood, let alone in neuropathic pain regulation. Limited literature indicates that HDAC9 can mediate the effect of neuron-elicited electrical activity[42,43] and knockdown of HDAC9 promotes dendritic growth in developing cortical neurons[44]. In the present study, our findings reveal that HDAC9 activity is required for trigeminal-mediated neuropathic pain behavior. We found that decreased HDAC9 expression associated with nerve injury increased acetylation at H3K18 in the miR-203a-3p gene promoter region and contributed to neuropathic pain behaviors. In support of our study, transcriptomic profiling of the rat TG by

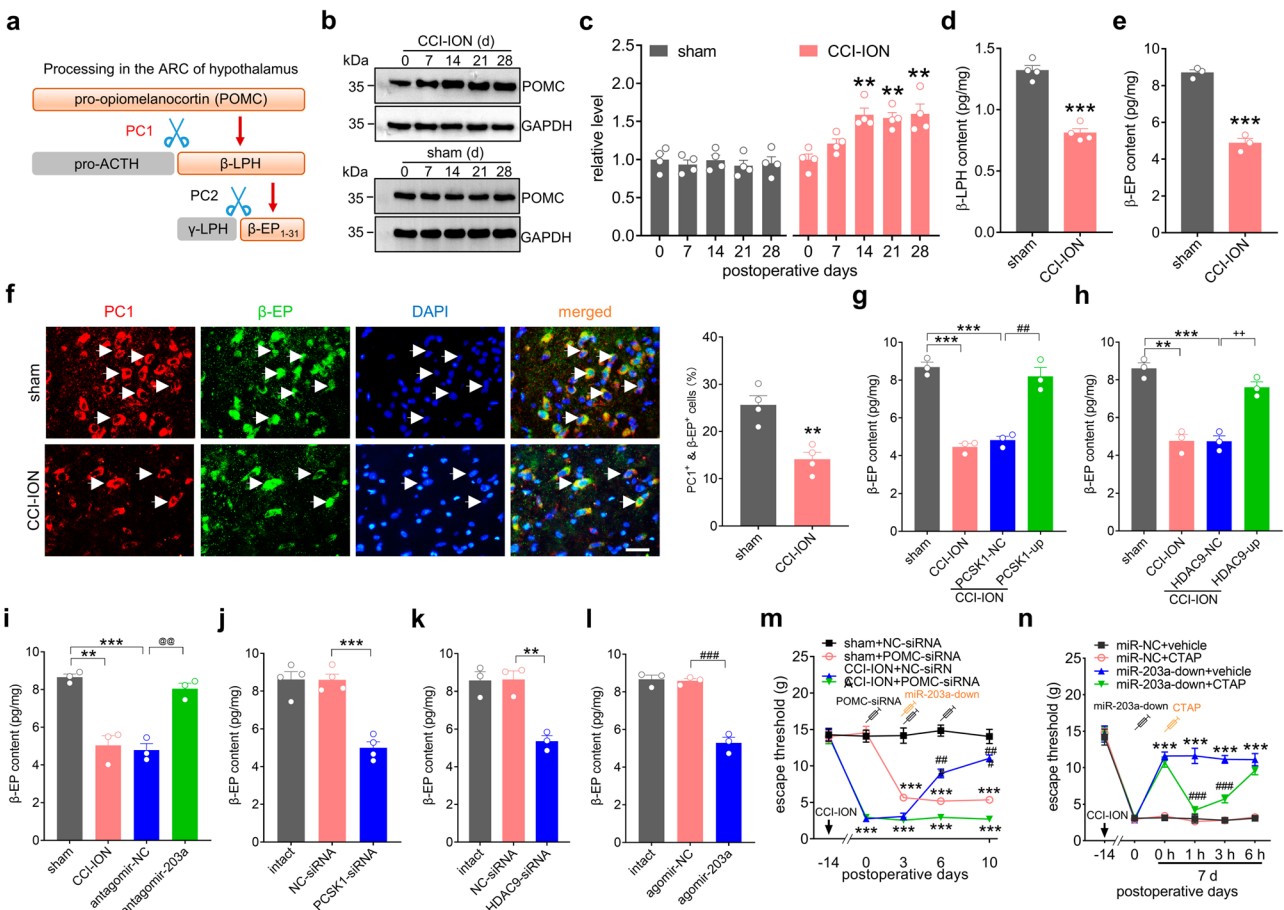

**Fig. 7 | Decreased β-EP mediates mechanical allodynia in CCI-ION rats.**
**a** Schematic diagram showing that PC1 cleaves POMC to β-LPH, and PC2 further cleaves β-LPH to β-EP. **b, c** Representative immunoblots (**b**) and bar charts (**c**) indicate the protein abundance of POMC in the ARC. **p < 0.01 versus POMC on Day 0, one-way ANOVA followed by Bonferroni's test. n = 8 rats per time point per group. **d, e** ELISA analysis of β-LPH (**d**, n = 4 rats per group, p = 0.0001) or β-EP (**e**, n = 6 rats per group, p = 0.0002). ***p < 0.001 versus sham, two-sided Student's t-test. **f** Immunostaining analysis of PC1 (red) with β-EP (green) and DAPI (blue) in the ARC. Bar charts show the percentage of both PC1+ and β-EP+ ARC neurons among total numbers of DAPI-labeled cells. Arrows show colocalization. **p = 0.0031 versus sham, two-sided Student's t-test. n = 4 rats per group. Scale bar, 25 μm. **g–i** Administration of PCSK1-up (**g**), HDAC9-up (**h**), or antagomir-203a (**i**) abolished the CCI-ION-induced β-EP decrease in the ARC. **p < 0.01, ***p < 0.001 versus sham, ##p = 0.0027 versus CCI-ION + PCSK1-NC, ++p = 0.0023 versus CCI-ION + HDAC9-

NC, @@p = 0.0021 versus CCI-ION + antagomir-NC, one-way ANOVA followed by Bonferroni's test. n = 6 rats per group. **j–l** Administration of PCSK1-siRNA (**j**), HDAC9-siRNA (**k**), or agomir-203a (**l**) decreased the content of β-EP in the ARC of intact rats. **p = 0.0037, ***p = 0.0002 versus NC-siRNA, ###p = 0.0005 versus agomir-NC, one-way ANOVA followed by Bonferroni's test. n = 6 rats per group. **m**, Evaluation of POMC-siRNA or NC-siRNA on the miR-203a-down-mediated attenuation of mechanical allodynia in CCI-ION rats. ***p < 0.001 versus sham + NC-siRNA, ###p < 0.001 versus CCI-ION + POMC-siRNA, two-way ANOVA followed by Bonferroni's test. n = 8 rats per group. **n** Intravenous administration of CTAP (0.1 mg/ml) completely abolished the miR-203a-down-induced alleviation of mechanical allodynia in CCI-ION rats. ***p < 0.001 versus miR-NC + vehicle at Day 0, ###p < 0.001 versus miR-203a-down + vehicle, two-way ANOVA followed by Bonferroni's test. n = 8 rats per group. All data are presented as mean values ± SEM.

RNA-seq analysis suggested that HDAC9 might be associated with referred hyperalgesia following masseter muscle inflammation[45]. In addition, although little is known regarding HADC9 in pain regulation and more in-depth experimental studies are needed, several studies using its broad spectrum chemical inhibitors have implicated a role for HDAC in nociceptive behaviors. For instance, histone deacetylase inhibitors, which promote histone acetylation, aggravate mechanical sensitization in incision-induced neuropathic pain models[46,47]. Some studies however, have shown contradictory results in that intrathecal injection of HDAC inhibitors attenuated pain hypersensitivity in models of traumatic nerve injury[48]. Nevertheless, it should be noted that these broad-spectrum HDAC inhibitors lack specificity, which limits the identification of specific HDACs involved in neuropathic pain following nerve injury. Although further studies are required to deduce the exact/complete repertoire of HDAC family members involved in pain regulation, the different regulatory effects of distinct classes of HDACs might vary, or even have opposite effects across

tissue types/species or pain models[46]. Indeed, in one study restoring HDAC2 in primary sensory neurons led to long-lasting relief of neuropathic pain[49], while in another, intrathecal injection with the HDAC2 siRNA, but not HDAC1, attenuated mechanical and thermal hypersensitivity[50].

The current study revealed that blockade of miR-203a-3p by miR-203a-down significantly attenuated neuropathic pain in both male and female rats. Specific pathways in spinal microglia and sensory macrophages have been shown involved in the sexual dimorphism of neuropathic pain[51,52]. Notably, sexual dimorphism may be limited to microglial and/or macrophages since the inhibition of pain-related signaling in neurons and astrocytes produced similar analgesia in both sexes[53,54]. Given that miR-203a-3p, downstream PC1, as well as its cleavage product β-EP, are expressed exclusively in ARC neurons, it accounts for the consistency of male and female rat pain symptoms observed in the present study. β-EP is an important endogenous opioid peptide primarily studied for its analgesic effects on pain perception in

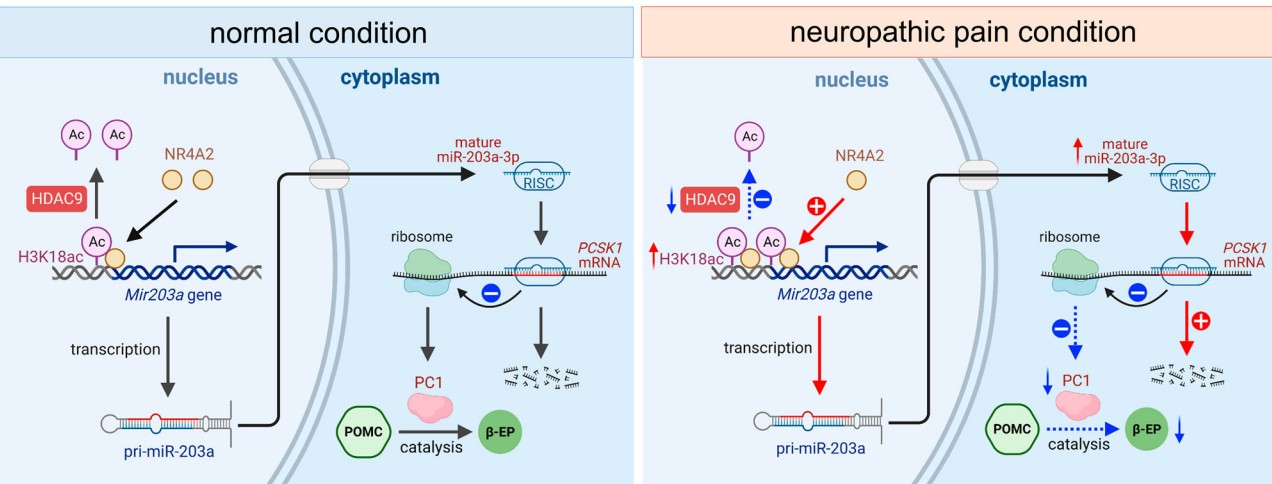

**Fig. 8 | Schematic diagrams illustrating the role and mechanism of miR-203a-3p in neuropathic pain.** The left panel demonstrates a low miR-203a-3p level in ARC neurons with deacetylated histones in the miR-203a-3p promoter region under normal conditions. The right panel reveals the mechanism underlying miR-203a-3p upregulation and β-EP decrease. First, peripheral nerve injury induces the downregulation of HDAC9, resulting in H3K18ac enrichment to facilitate the binding of NR4A2 in the *miR-203a-3p* promoter region, which leads to an upregulation in the expression of miR-203a-3p. Next, the functional RNA-induced silencing complex guides the mature miR-203a-3p to recognize and bind the complementary PCSK1 mRNA sequence within the 3'-UTR, leading to the degradation of target mRNA. Finally, the upregulation of miR-203a-3p in nerve injury causes a reduction in PC1 expression, subsequently inducing a substantial and persistent decrease in the content of β-EP in ARC neurons and eventually leading to neuropathic pain. Created with BioRender.com.

both the central and peripheral nervous systems[55]. β-EP-synthesizing neurons are predominantly located in the ARC, which play pivotal roles in nociceptive behaviors[56,57]. The availability of β-EP is mainly determined by the expression levels of POMC and subsequent proteolytic processing by a combination of PC1 (cleaving POMC to β-LPH) and PC2 (cleaving β-LPH to β-EP)[23,24]. In the present study, the expression of PC2 protein was unaffected in the rat ARC after CCI-ION, while both PC1 and its cleavage product β-LPH, the initial hydrolysate produced by POMC, were significantly decreased, indicating that PC1 is the determinant convertase that decreases β-EP levels. In the ARC of CCI-ION rats, since POMC synthesis (mRNA level) remained unchanged, a decrease in PC1 led to less cleavage of POMC, consequently causing an accumulation of POMC. As such, our findings strongly suggest that miR-203a-3p-mediated PC1 downregulation induces mechanical hyperalgesia through inhibition of β-EP generation in rat ARC neurons following nerve injury. As miR-203a-3p affects the expression of additional genes further studies are required concerning factors relevant to both nociceptive and other pathways[58]. Consistent with our current findings, previous studies have shown that reduced levels of β-EP may contribute to the chronic pain state experienced by trigeminal neuralgia patients[59]. Moreover, patients with chronic neuropathic pain due to trauma or surgery exhibited decreased levels of β-EP in their cerebrospinal fluid, suggesting a defective top-down modulation in the chronic neuropathic pain mechanism[12]. Notably, angiotensin II type 2 receptor-deficient mice exhibit increased sensitivity to pain and decreased levels of ARC β-EP[60]. Contrastingly, studies have shown that the levels of β-EP in the ARC are not necessarily decreased and even elevated in inflammatory pain models. For instance, applying a brain microdialysis method, Zangen et al. demonstrated that a peripheral noxious stimulus after formalin injection increased the extracellular level of β-EP in the rat ARC[9]. Additionally, the release of endogenous opioid peptides was significantly elevated in a carrageenan-induced inflammatory pain model[7]. Although these discrepancies have yet to be more fully explained, the various observations suggest that different modes of pain sensitivity (inflammatory pain *vs.* neuropathic pain) may not necessarily involve similar β-EP changes. In addition, different sampling, including plasma, tissue slice preparations and extracellular fluid, used to determine the release of β-EP in the brain vary by

analytical approach. Indeed, an earlier study found that in response to painful stimuli, β-EP rose in blood plasma but fell in hypothalamic tissue from the same animal[60,61]. Moreover, it should be noted that nociceptive stimuli promote the release of β-EP into the extracellular fluid of the hypothalamic ARC[9], which might account for the reduction in tissue β-EP stores as described previously[61]; this was also consistently observed in our present study. β-EP is also known to be involved in stress[62]. For instance, physical and fearing-inducing psychological stressors stimulate β-EP release in the ARC[62]. In addition, a recent study showed that chronic restraint stress decreases the excitability of hypothalamic POMC neurons[63], together suggesting a potential role of stress in β-EP regulation. Here, we found that nerve injury upregulated miR-203a-3p in the ARC, resulting in a decrease of β-EP. Interestingly, it has been shown that chronic stress did not induce changes of miR-203a-3p expression in the HIP and medial prefrontal cortex[64]. Additionally, injection of chemical stressors did not elicit changes of β-EP expression in the ARC[62]. Although stressors with different properties are processed differently in the brain[62], it can be speculated that increased miR-203a-3p in the ARC was less likely due to the stress response induced by CCI-ION. Consistent with this notion, previous studies have revealed that chronic stress had no influence on mechanical threshold[65]. Nevertheless, chronic pain and stress are two complex states, and they may mutually exacerbate one another in conditions of comorbidity. Further investigation is necessary to examine the role of stress in trigeminal-mediated neuropathic pain; however, we believe it is beyond the scope of the current study.

In summary, our findings offer insights into epigenetic regulatory mechanisms underlying endogenous β-EP synthesis in ARC neurons involved in the pathogenesis of neuropathic pain. We demonstrate that nerve injuries cause a decrease in HDAC9 expression resulting in H3K18ac accumulation that facilitates the binding of NR4A2 to the miR-203a-3p gene promoter. Upregulated miR-203a-3p leads to a reduction in PC1 expression and subsequently induces a substantial and persistent decrease in β-EP in ARC neurons resulting in mechanical allodynia. The identification of epigenetic modifiers such as HDAC9 and miRNA-203a-3p in regulating PC1 in chronic neuropathic pain may provide potential targets for drug development and the treatment of this disorder. Nevertheless, the administration of epigenetic drugs

necessitates caution, as it can be akin to a double-edged sword, potentially yielding unexpected side effects. It is imperative to acknowledge that the interventions of epigenetic modifiers such as NR4A2, H3K18ac, and miR-203a-3p may trigger the activation of genes associated with cognitive disorders[66] and proinflammatory genes[67,68]. Given the intricate and diverse nature of epigenetic modifications and their combinations, a careful assessment of epigenetic treatments is essential. The cellular specificity and safety and of miR-203a-3p and its modifiers will require further evaluation to address any potential off target activity. Moreover, it is known that the endogenous endorphinergic system is suppressed when potent exogenous opioid compounds are given at an excessive amount over a long period of time, which leads to opioid tolerance[69]. Whereas the continuous presence of endogenous opioids might preserve signaling of μ-receptors and thereby counteracts the development of tolerance[70]. Therefore, it is tempting to propose that direct promotion of endogenous β-EP synthesis may provide an avenue for the development of analgesia without tolerance.

## Methods

### Animal model

Animals were purchased from Shanghai SLAC Laboratory Animal Co., Ltd. (Shanghai, China) and maintained at the Soochow University Animal Facility. All animal procedures were approved by the Animal Care and Use Committee of Soochow University strictly in accordance with the National Institutes of Health (NIH) guidelines for animal research and the International Association for the Study of Pain. Adult Sprague–Dawley rats (male and female, 8 – 10 weeks) were housed (three rats per cage with soft bedding) in specific temperature- and humidity-controlled facilities on a 12/12 h light-dark cycle with food and water available *ad libitum*. Rats used in the study referred to the male ones unless otherwise specified. Every effort was made to minimize suffering and the number of animals used. Before any experiments, animals were adapted to the test environment for three days. The chronic constriction injury to the left infraorbital nerve (CCI-ION) model was prepared as a trigeminal neuropathic pain model, which was established by an intraoral approach[17,71] with some modifications. Briefly, rats were deeply anesthetized with isoflurane during surgical procedures, and an incision of approximately 1 cm was made lengthwise along the gingivobuccal margin in the buccal mucosa of the beginning just proximal to the left first upper molar. The adhering tissue surrounding the incision was separated by blunt dissection until the ION was exposed. The ION was raised with glass rods, and two sutures (4-0 silk) separated 2 to 3 mm apart were tied loosely around it. The same surgical procedure was performed on sham-operated rats, and only the unilateral nerve was exposed without ligation. To produce a spinal nerve ligation (SNL), the L5 transverse process was removed to expose the L4 and L5 spinal nerves. The L5 spinal nerve was then isolated and tightly ligated with 6-0 silk thread. In sham-operated rats, the L5 spinal nerve was similarly exposed, but without ligation.

### Behavioral tests

All of the behavioral experiments were performed by investigators blinded to the treatment. Orofacial behavioral tests were accurately performed 1 day before and at scheduled times after any surgeries. Each rat was individually placed in a plastic cage, and a 2 h environmental adaptation period was set before testing. Stimuli were applied to the orofacial skin near the center of the left vibrissal pad innervated by the ION. A series of von Frey filaments with increasing strength from 0.008 g to 15 g (Stoelting) was consecutively applied at least 3 times, 2 s apart[17,72–74]. To keep the head of rats from tissue damage and turning under excessive mechanical force, the cutoff was set to the 15 g filament. The escape threshold was tested with a graded series of calibrated *von* Frey filaments in a sequential ascending force until positive nociceptive behavior occurred. As described by Vos et al.[74]. and in our previous study[17], a positive nociceptive behavior was noted when at least one of the following behaviors were triggered: 1) a brisk head withdrawal in which the rat pulls briskly backward, usually followed by a consecutive series of at least 3 face-wash strokes directed to the stimulated facial area; 2) an attack reaction in which the rat actively attacks the filament or makes biting and grabbing movements; and 3) an escape reaction in which the rat passively moves its body away from the filament and assumes a crouching position near the cage wall to avoid further mechanical stimulation. The minimal force to induce positive nociceptive behavior through *von* Frey filaments was defined as the mechanical escape threshold. For testing mechanical sensitivity of SNL rats, animals were put in a plastic box (11 × 13 × 24 cm) on an elevated metal mesh floor and allowed 30 min for habituation. Mechanical pain sensitivity was determined by measuring paw withdrawal threshold (PWT) of rats via *von* Frey filaments (Stoelting, 0.008 g to 15 g)[75]. The 50% withdrawal threshold was determined using the up-down method of Dixon[73].

### Real-time quantitative PCR (RT–qPCR)

The rats were euthanized with an overdose of sodium pentobarbital (150 mg/kg) by intraperitoneal injection (i.p.) and decapitated, and the entire brain was quickly removed and placed in a mold on ice where it could be sliced into 2.5 mm coronal sections. The ARC tissue samples were rapidly dissected from the slices between Zeitgeber time (ZT) 6 and ZT8. After extraction of the total RNA from the rat ARC using Takara RNAiso Plus, the quality and RNA concentrations of samples were assessed from absorbance measurements using a NanoDrop spectrophotometer (NanoDrop One, ThermoFisher Scientific). The Hairpin-it™ MicroRNA Quantitation PCR kit was purchased from GenePharma, and the PrimeScript RT Master Mix was purchased from Takara. RT–qPCR was conducted according to the manufacturer's protocol. Purified RNAs (1 μg) were reverse-transcribed into cDNA using the following protocol: 16 °C for 30 min (10 min for mRNA), 42 °C for 45 min (15 min for mRNA) and 85 °C for 10 min (2 min for mRNA). Each experiment was performed with at least three biological replicates. qPCR analysis was conducted using the Light-Cycler 96 system (Roche). The PCR quantification of miRNA was performed using the $2-^{\Delta\Delta Cq}$ method against the internal control U6 for normalization. mRNA expression was determined with SYBR Green PCR Master Mix (Takara). Relative expression of mRNA was calculated using the $2^{\Delta\Delta Cq}$ method against GAPDH for normalization. The specificity of the amplified PCR product was verified by agarose gel electrophoresis and melting-curve analysis. The RT–qPCR primers are summarized in Tables S1, S2.

### Drug application and intra-ARC injection

Stereotaxic intracerebral cannula implantation for drug/reagent administration was conducted[76]. In brief, the rat was anesthetized with isoflurane and mounted steadily on a stereotaxic instrument. ARC coordinates were determined with reference to the rat brain in stereotaxic coordinates prepared by George Paxinos & Charles Watson. A bioclean stainless-steel guide cannula was positioned 2.8 mm dorsal to the ARC [anterior (+) or posterior (−) to Bregma (AP), −4.0 mm; lateral to midline (L), unilateral 0.5 mm; ventral to the surface of skull (V), -9.8 mm] and fixed firmly to the skull by dental acrylic. Rats were bilaterally injected using a twenty-gauge microinjection needle (Hamilton) with 0.5 μl of reagents. Injections were conducted within a period of 5 min, and the needle was kept within the guide cannula for at least 1 min before removal. The injection sites within the ARC were histologically verified afterward by injecting blue dye which was mixed with the injection solutions so that accurate targeting of the ARC could be monitored. As shown in Fig. S25, blue color appeared within the ARC area with no apparent spreading of the injected reagents into the surrounding areas, ensuring the effects of reagents being specific to ARC without any spreading effect on the neighboring brain regions. 2'-O-methyl-modified and 5'-cholesteryl-modified small interfering RNA (siRNA) for HDAC9 (HDAC9-siRNA), PCSK1 (PCSK1-siRNA) and POMC

(POMC-siRNA) (RiboBio Biological Technology, China), agomir-203a-3p (GenePharma, China), antagomir-203a-3p (GenePharma, China), or the relevant scrambled controls (NC-siRNA, agomir-NC, or antagomir-NC), labeled with 6-carboxyfluorescein (6-FAM), were diluted in diethyl pyrocarbonate (DEPC)-treated water at a concentration of 50 µM. Agomir, antagomir, or siRNA sequences were BLAST against the rat GenBank (https://blast.ncbi.nlm.nih.gov/) to exclude the presence of multiple target sequences in the rat genome, and are summarized in Table S3. All of the neuron-specific human synapsin 1 gene promoter (hSyn) combinatorial lentiviral vectors carrying the enhanced green fluorescent protein (eGFP) gene, including lenti-hSyn-HDAC9-up (HDAC9-up), lenti-hSyn-PCSK1-up (PCSK1-up), lenti-hSyn-miR-203a-3p-up (miR-203a-up), lenti-hSyn-miR-203a-3p-antisense (miR-203a-down) and the relevant negative controls, were purchased from GenePharma. The viral titer was at least $1 \times 10^9$ TU/ml. The highly selective µ-opioid receptor antagonist CTAP was obtained from MedChemExpress (MCE) and intravenously injected at 0.5 mg/kg.

## Immunoblot analysis

The rats were euthanized with an overdose of sodium pentobarbital (150 mg/kg, i.p.) and decapitated, and the entire brain was quickly removed and placed in a mold on ice where it could be sliced into 2.5 mm coronal sections. The ARC tissue samples were collected between ZT6 and ZT8 for immunoblot analysis[17,77]. Proteins (30 µg) were separated on 10% (for HDAC9, NR4A2, POMC, PC1 and PC2 proteins) or 12% (for H3K9ac, H3K14ac, H3K18ac, H3K27ac and H4ac proteins) SDS–PAGE gels before being transferred to polyvinylidene difluoride (PVDF) membranes (Merck Millipore). After blocking the membranes with 5% nonfat milk, bound proteins were exposed to primary monoclonal antibodies against the following: H3K9ac (rabbit, 1:1000, Merck Millipore), H3K14ac (rabbit, 1:1000, Merck Millipore), H3K18ac (rabbit, 1:1000, Abcam), H3K27ac (rabbit, 1:1000, Merck Millipore), H4ac (rabbit, 1:800, Abcam), H3 (rabbit, 1:100, Abcam), HDAC9 (rabbit, 1:1000, Abcam), NR4A2 (mouse, 1:1000, Abcam), POMC (rabbit, 1:2000, Abcam), PC1 (rabbit, 1:1000, Abcam), PC2 (rabbit, 1:1000, Abcam), and glyceraldehyde 3-phosphate dehydrogenase (GAPDH, mouse, 1:1000, Cell Signaling Technology). The membrane was washed three times with TBST, followed by incubation with a 1:8000 dilution of goat anti-mouse or goat anti-rabbit horseradish peroxidase secondary antibody (R&D Systems). The membranes blotted for acetylation-modified histone were stripped with Western Blot Stripping Buffer (Beyotime) for 10 min, washed three times with TBST, blocked again with 5% nonfat milk, and re-probed with primary antibodies against total H3 proteins (rabbit, 1:100, Abcam). The blots were detected and visualized by the ChemiDoc XRS system (Bio-Rad Laboratories) after being incubated with SuperSignal West Pico Plus chemiluminescent substrate kit from ThermoFisher Scientific. The intensity of each immunoreactive protein band was analyzed by densitometry using Quantity One software. For calculating relative changes induced by any treatment, the ratio of the band intensity versus that of GAPDH or H3 was calculated and then normalized to the ratio in the control group.

## Immunofluorescent staining

Immunofluorescent staining was performed to detect the subcellular localization[17,78]. All rats were deeply anesthetized with sodium pentobarbital (100 mg/kg, i.p.), followed by transcardial perfusions. In brief, the brain was fixed in 4% paraformaldehyde, washed three times with PBS, and dehydrated in 15% and 30% sucrose in PBS. After embedding in O.C.T. compound (Leica), samples were cut in a coronal plane with a thickness of approximately 12 µm using a Leica CM1950 cryostat. The sections were permeabilized with 0.15% (w/v) Triton X-100 for 15 min, briefly washed twice with PBS, and blocked in 10% normal goat serum for 1 h at room temperature. The sections were then incubated overnight at 4 °C with primary antibodies against c-Fos

(mouse, 1:300, Abcam), HDAC9 (rabbit, 1:300, Abcam), H3K18ac (rabbit, 1:300, Abcam), NR4A2 (mouse, 1:300, Abcam), PC1 (rabbit, 1:300, Abcam), NeuN (mouse, 1: 300, Merck Millipore), GFAP (mouse, 1:300, Cell Signaling Technology), CD11b/c (mouse, 1:200, Abcam), and β-endorphin (mouse, 1:200, Abcam). After washing with PBS, the sections were incubated with Alexa Fluor® 555-conjugated goat anti-rabbit IgG (1:300, Cell Signaling Technology), Alexa Fluor® 488-conjugated goat anti-rabbit IgG (1:300, Cell Signaling Technology) or Alexa Fluor® 488-conjugated goat anti-mouse IgG (1:300, Cell Signaling Technology) for 2 h. Images were acquired on an upright fluorescence microscope (104 C, Nikon) equipped with a CCD camera (Photometrics CoolSNAP HQ2).

## Fluorescence in situ hybridization (FISH)

The locked nucleic acid (LNA) probes with complementarities to miR-203a-3p labeled with 5' and 3'-digoxin and the relevant negative control probes were synthesized by Exon Biotechnology (Guangzhou). The brain sections were washed with DEPC-PBS and incubated in acetylation buffer for fifteen minutes. After rinsing three times in DEPC-treated PBS, sections were prehybridized in hybridization buffer (50% formamide, 5×Denhardt's solution, dextran sulfate, saline-sodium citrate, and 200 µg/ml yeast RNA) at 55 °C for 1 h to lower the background. The rno-miR-203a-3p miRCURY LNA detection probe (5'-digoxin−CTAGTGGTCCTAAACATTTCAC−digoxin-3') and its negative control were denatured for 3 min and stabilized for 2 min at 4 °C. After incubation with the denatured hybridization probe, the sections were rinsed in washing buffer (saline-sodium citrate and formamide), blocked in blocking buffer (1% BSA in DEPC-treated PBS), and incubated with Alexa Fluor 555-labeled anti-digoxin antibody (1:300 dilution). For FISH analysis combined with immunofluorescent labeling, sections were further processed for immunostaining with NeuN (mouse, 1: 300, Millipore), GFAP (mouse, 1:300, Cell Signaling Technology), CD11b/c (mouse, 1:200, Abcam), PC1 (rabbit, 1:300, Abcam), NR4A2 (rabbit, 1:300, Abcam), and H3K18ac (rabbit, 1:300, Abcam). After washing with PBS, the sections were incubated with DyLight® 488-conjugated goat anti-rabbit IgG (1:400, Abcam) or DyLight® 488-conjugated goat anti-mouse IgG (1:400, Abcam) for 2 h and photographed under a fluorescence microscope equipped with a CCD camera (Photometrics CoolSNAP HQ2).

## Luciferase reporter assay

Different truncated regions of the *miR-203a-3p* gene promoter (fragments 1 to 5) were amplified using specific primers and separately inserted into the pGL3-basic luciferase reporter plasmid (Promega) with the endonucleases Hind III and XhoI to generate a series of plasmids. The public database JASPAR (http://jaspar.genereg.net/) was applied to predict the potential binding sites of transcription factors, including SP1, NFKB1, NFATC2, NR4A2 and SOX10, on the 351-bp long region (ΔF, bp −1275 to bp -1625) in the *miR-203a-3p* gene promoter. Sequences containing mutant binding sites for SP1 (mut-SP1), NFKB1 (mut-NFKB1), NFATC2 (mut-NFATC2), NR4A2 (mut-NR4A2) or SOX10 (mut-SOX10) in the *miR-203a-3p* gene promotor region were cloned into the pGL3-basic plasmid. The wild-type full-length 351 bp fragment (ΔF) derived from the *miR-203a-3p* gene promotor (miR-203a-3p-wt) was utilized as a control group. The plasmids containing firefly luciferase followed by the wild-type PCSK1 3'-UTR (PCSK1-wt) or the mutant PCSK1 3'-UTR (PCSK1-mut) were also cloned into the pGL3 vectors (Azenta Life Science). All wild-type and mutant constructs were obtained from Azenta Life Science (Suzhou) and confirmed by Sanger sequencing. The Renilla luciferase vector pGMR-TK (Promega) cotransfected with pGL3-basic plasmid (control) was used for data normalization. PC12 cells and HEK293 cells used in this study were obtained from Cell Bank of Type Culture Collection of the Chinese Academy of Sciences (Shanghai Institute of Cell Biology). For transfection, approximately $1 \times 10^5$ cells were seeded into each well of a

24-well plate and allowed to adhere overnight. PCSK1-wt-3′-UTR, PCSK1-mut-3′-UTR plasmid (500 ng), pGMR-TK (100 ng), and miR-203a-3p mimics (100 ng) (or the negative control) were transfected into HEK293 cells using Lipofectamine 6000™ (Invitrogen). Similarly, miR-203a-wt (500 ng), mut-SP1 (500 ng), mut-NFKB1 (500 ng), mut-NFATC2 (500 ng), mut-NR4A2 (500 ng), mut-SOX10 (500 ng), and pGMR-TK (100 ng) were transfected into PC12 cells using Lipofectamine 6000™ (Invitrogen). Forty-eight hours after transfection, both firefly and Renilla luciferase activities were consecutively determined with a Dual-Glo Luciferase Assay System (Promega) in a TD-20/20 Luminometer (Turner BioSystems, USA). All experiments were repeated independently at least three times and conducted in triplicate (technical replicates) for each group. For data processing, the relative ratio was calculated by normalizing firefly luciferase activity to Renilla activity and normalized to the ratio in the control group.

### Chromatin immunoprecipitation (ChIP)-PCR

ChIP was performed using a SimpleChIP plus Enzymatic Chromatin IP kit (Cell Signaling Technology) according to the manufacturer's protocol[17]. To crosslink the proteins to DNA, ARC lysates were treated with PBS containing 1.5% formaldehyde and 0.5% complete protease inhibitor cocktail, followed by quenching with glycine. Tissues were further ground using a mechanical tissue homogenizer, and precipitated nuclear tissues were collected by centrifugation. Subsequently, genomic DNA was digested into fragments of approximately 150–500 bp in length with micrococcal nuclease and transferred into ChIP buffer containing 0.5% protease inhibitor cocktail. After sonication and centrifugation, 2% of the supernatant was chosen as the input DNA control. The remaining supernatant was incubated with 2 μg of antibodies against H3K18ac (Abcam) or NR4A2 (Abcam) to pull down chromatin. The supernatant incubated with IgG (Cell Signaling Technology) was used as a negative control. ChIP-grade protein G magnetic beads (Cell Signaling Technology) were used to capture protein-chromatin complexes. After washing, the protein-chromatin complexes were eluted with ChIP elution buffer (0.1 M NaHCO3 and 1% SDS) and decrosslinked by the addition of proteinase K. The DNA product was purified using affinity purification columns (Cell Signaling Technology) and collected. qPCR analyses were subsequently performed to quantify the ChIP-enriched DNA using the following primers specific for the NR4A2-binding region (forward 5′-TTCGTCACGCCCTCTGTTTT-3′ and reverse 5′-TATGGGCCCCTATGGCTCTC-3′). The enrichment level was calculated and normalized to the input samples. The PCR product was subsequently separated by electrophoresis on a 2% agarose gel (Sigma–Aldrich).

### Enzyme linked immunosorbent assay (ELISA)

ELISA was performed following the manufacturer's protocol[78]. Briefly, both the rat β-EP ELISA kit and the rat β-LPH ELISA kit were purchased from BBI Life Sciences. The in-house IgG ELISAs utilized anti-rat IgG to coat the 96-well plate. The combined β-EP or β-LPH antigens from ARC homogenate or standards (50 μl) were incubated within the wells for 45 min. After washing, the biotin-conjugated β-EP or β-LPH antibody working solution, HRP-conjugated streptavidin working solution, and TMB substrate were added to the specimen and incubated within the wells for 30 min, consecutively. The stop solution (2 M HCl) was added to terminate the reaction, and the optical density (OD) values were read at A450 nm using a TD-20/20 Luminometer (Turner BioSystems, USA).

### Bioinformatics analysis

Public databases, including TargetScan (www.targetscan.org/), miR-anda (www.microrna.org/microrna), miRWalk (mirwalk.umm.uni-heidelberg.de/) and miRDB (www.mirdb.org/), were used to predict the potential targets of miR-203a-3p. JASPAR (https://jaspar.genereg.net/) was employed to analyze the potential transcription factors involved in

the transcriptional regulation of miR-203a-3p. MethPrimer (www.urogene.org/cgi-bin/methprimer) was used to identify the CpG island in the *miR-203a-3p* gene promoter region.

### Patient population and CSF collection

All patients gave written informed consent to sample collection and data analysis prior to study entry. All CSF samples were obtained with informed consent compliance with the Ethical Committee of the First Affiliated Hospital of Soochow University. A total of 7 healthy subjects and 9 patients with TN were recruited for this study, and they had not suffered from any other neurological disorder (Table S4). The reference standard clinical diagnosis for TN could be based on defined criteria or judgment of 1 or more experienced doctors (neurologist, pain specialist). Samples of fasting CSF (3–5 ml) were obtained in the morning by standard lumbar puncture, collected in 10 ml polypropylene tubes, and biochemical parameters were within normal range. Following centrifugation, CSF samples were aliquoted and stored at −80 °C until analysis. miRNAs were isolated from 200 μl of human CSF using the miRNeasy Serum/Plasma Kit (Qiagen) according to the manufacturer's instructions.

### Data analysis

All data are presented as mean values ± SEM. Data acquisition and statistical analysis were performed using Microsoft Excel and Prism 8.0 (GraphPad Software). No statistical methods were used to predetermine sample sizes, and no randomization algorithm was used, although rats were randomly assigned to experimental conditions. One-way ANOVA with Bonferroni's *post hoc* test to compare three or more groups or Student's *t* test to compare two groups was used for statistical analysis as appropriate. Two-way repeated-measures ANOVA followed by Bonferroni's multiple comparisons test with a *post hoc* test was used to analyze behavioral data with two independent variables. Differences with $p$ values < 0.05 were considered to be statistically significant.

### Reporting summary

Further information on research design is available in the Nature Portfolio Reporting Summary linked to this article.

## Data availability

The raw sequencing data generated in this study have been deposited in the National Center for Biotechnology Information Gene Expression Omnibus (GEO) under the accession number GSE216965. All other study data are included in the article and *SI Appendix*. Source data are provided with this paper.

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

## Acknowledgements

This study was supported by the National Natural Science Foundation of China (82371218, 82271245, and 82071236), the Natural Science Foundation of Jiangsu Province (BK20211073), the Science and Technology Bureau of Suzhou (SYS2020129), the Jiangsu Key Laboratory of Neuropsychiatric Diseases (BM2013003), the Project of State Key Laboratory of Radiation Medicine and Protection (GZK1202223), Clinical Research Center of Neurological Disease (ND2022B03), Postgraduate Research & Practice Innovation Program of Jiangsu Province (KYCX23_3260), Research Fund from MOE Key Laboratory of Geriatric Diseases and Immunology, and a project funded by the Priority Academic Program Development of Jiangsu Higher Education Institutions.

## Author contributions

J.T. as the corresponding author conceived the project, supervised all experiments, and funded the work. Y.T., Y.Z., X.J., N.H., H.L., R.Q., Z.H., Y.S., D.J., T.P.S., and X.J. contributed toward the development and execution of the project, each making substantial contributions toward this work, including design, acquisition, analysis, or interpretation of the data presented. The manuscript was drafted by Y.T., Y.Z., and J.T.; All authors revised the article critically for intellectual content. All authors read and approved the final manuscript.

## Competing interests

The authors declare no competing interests.
