## [Peer Review File · Nature Communications]

Epigenetic regulation of beta-endorphin synthesis in hypothalamic arcuate nucleus neurons modulates neuropathic pain in a rodent pain modelREVIEWER COMMENTS

Reviewer #1 (Remarks to the Author):

The manuscript by Tao and colleagues identify a novel epigenetic mechanisms underlying endogenous β -EP synthesis driven by microRNA regulation in ARC neurons involved in the pathogenesis of neuropathic pain. The manuscript is well-written and experiments are well designed. Results obtained in behavioral and molecular biology experiments are very consistent. The identification of epigenetic modifiers, such as microRNA-203a-3p, provides potential targets for development of chronic neuropathic pain.

However, there are some concerns as follows:

One concern is that whether modulation of these epigenetic modifiers would lead to any potential adverse effects on the body. This aspect could be speculated in the discussion.

We note that only male rats were used in the manuscript, I recommend either being very transparent about this limitation in the manuscript or replicating one or two of key findings using female rats.

As a chronic neuropathic pain model, the CCI-ION model was used here. If the authors could replicate one or two key findings using 1-2 other neuropathic pain models, the evidence supporting these epigenetic modifiers as potential targets would be stronger.

It would be better to provide the qualitative analysis data showing statistic counting of the fluorescent expression neurons, for example, Fos-positive neurons in Figure 1B, as well as counting the percentage of miR-203a-3p-positive cells colocalized with neuronal markers such as in Figure 1K, 3H.

In line 110, the authors found that miR-203a-3p in the ARC was comparatively higher relative to other brain areas, but the ARC was not the highest area. As shown in Figure 1F, the highest area was NAc, and HIP was in comparable level with ARC. The authors could explain the potential effects of these areas.

Another concern is that whether the antagomir or agomir used in this study was specific to manipulate miR-203a-3p. If not, it could not exclude the effect of other miRNAs. The authors used neuron-specific promoter lenti-hSyn-miR-203a-up to increase miR-203a expression. These are sufficient experiments to demonstrate the miR-203a-3p played a pivotal role in the neuropathic pain. It would be better if the authors could perform the necessary experiments with neuron-specific manipulation to decrease or block the expression of miR-203a in CCI-ION mice and test whether the manipulation would induce analgesia.

Minor details:

Grammar mistakes, for example in line 109

Figure 1B: "ham" should be "sham"

Figure 1C: less of description of the color bar

Figure 1D: inconsistent in star character size

Figure 4K: the quality of the blots was not good. It would be better if the authors could provide better images.

Reviewer #2 (Remarks to the Author):

Central to this manuscript is the elucidation of the role of miR-203a-3p that is overexpressed in the hypothalamic ARC neurons, much lower in astrocytes or microglia after chronic constriction ligation of the infraorbital nerve to model neuropathic pain in the rat. The manuscript then described the transcriptional regulation of miR-203a-3p expression by transcriptional factor NR4A2 via epigenetic regulation of acetylation of the miR-203a-3p promoter by HDAC9, while NR4A2 protein level was similar between sham and CCI-ION rats. After, determining the upstream regulation of miR-203a-3p transcript expression, the authors examined the downstream target of the ncRNA and found that the PCSK1 gene that codes for PC1 was downregulated by it. The authors then determined the potential link between PC1 downregulation and b-EP production from POMC in the ARC neurons and related their work to mechanical allodynia.

Overall, the experiments were well designed and the rationale for selecting which transcriptional factor or acetylase or deacetylase to focus on was reasonable and tested in well controlled experiments. Could the authors address the following comments and suggestions with the aim to improve the clarity and depth of the discussion:

1. High levels of miRNA 203a-3p were also detected in brain regions like hippocampus and NAc (Fig 1F). As NAc also plays an important role in pain processing, is the current mechanism also applicable to NAc? Was miRNA 203a-3p also increased in NAc post CCI and could the authors comment on this in Discussion?
2. POMC is known to be circadian rhythmically regulated. As such, what ZT time were all the samples collected for detection of POMC and PC1 levels?
3. Could the authors comment on the increased level of POMC in the CCI-ION rats. What role does an increase in ARC POMC serve as later its cleavage will be decreased via an epigenetic mechanism that downregulates PC1, and the eventual pathway will subsequently lead to less β -EP being generated? Is this a compensatory or neutral process?
4. β -EP is also known to be involved in stress. A recent study has also shown that chronic restrained stress decreased the excitability of POMC neurons. Could the increase of miRNA 203a-3p also be due to the stress response induced by CCI?
5. In all the experiments with injections of siRNA or agomir/antagomir-203a to ARC, how did the authors ensure the effect is specific to ARC and that any spreading effect did not affect the neighbouring brain regions?
6. The authors showed that down-regulation of miR-203a-3p would reduce nociceptive behaviours in the CCI-ION rats. If using a pharmacological method to relief pain, would the level of miR203a-3p change

in response to the drug?

7. When using POMC-siRNA, multiple injections were performed (Fig 7M). What is the reason for injecting three times? For the same Fig 7M, is the injection of miR203a-down applied to all groups or the CCI-ION rats only? Please clarify in the legend.

Minor Comments:

1. Title: would authors consider using “modulates” instead of “controls”, as epigenetic phenomenon is dynamic.
2. Line 35: suggest changing “...increased acetylation” to “decreased deacetylation”, as this is a more intuitive description.
3. Line 37: what does “upstream” mean in the sentence. Please be clear.
4. Line 37: The sentence beginning with “The increased miR-203a-3p.....” is structured awkwardly and I suggest the authors rephrase.
5. Lines 40 and 466: change “revenue” to “an avenue”
6. Line 46: should it not be ... lesions and diseases of somatosensory nerves rather than nervous system?
7. Line 50: suggest to use the word “modulation” instead of “pathogenesis”.
8. Line 80: alteration instead of alternation

Reviewer #3 (Remarks to the Author):

Tao et al identify a new epigenetic pathway regulating beta-endorphin regulation in neuropathic pain. The authors do a good job of systematically following the pathway from regulatory microRNAs all the way to beta-endorphin regulation in a model of neuropathic pain. They use a series of straight-forward approaches to show that miR-203a-3p is upregulated in a model of chronic pain and in human patients, and that this miR promotes nociceptive behavior by regulating levels PCSK1. In addition, they show that miR-203a-3p expression is regulated by the transcription factor NR4A2, whose binding is regulated by HDAC9. Overall, I think this is a comprehensive series of experiments that elucidates a new pathway of pain regulation. I have several minor comments.

Minor comments

Figure 2E. Please add zoomed out image of ARC to see extent/spread of lentiviral infection.

Figure 3H. How many rats were used for IHC/in situ?

Figure 4A. what were the western blots normalized to? H3 mods are all about the same weight as total H3 and since blots were labeled with chemiluminescence, it's not clear how they could both be run on the same membrane to allow for normalization. Were the blots stripped? If so, what was the stripping protocol and in what order were the antibodies incubated? In other figures, GAPDH is used to normalize, but staining usually shows just the GAPDH band, so it's also not clear if these were done on

the same membrane?

Figure 4C. Please clarify how CHIP quantification was normalized. Figure captions say that data are normalized to input, but it looks like they may also be normalizing to sham?

Figure 6C. The authors report a correlation between PCSK1 expression and miR-203a expression. Why are there 2 clusters? If this reflects a group difference between CCI-ION vs control, data should be shown as bar graph, not correlation. Figure caption states that this figure shows protein abundance at day 0, 7, 14, 21, and 28, so the correlation may reflect differences between miR-203a-3p and PC1 levels over this time period? This is not clear in the text. Either way, use of correlations to describe 2 distinct data clusters seems inappropriate

Figure 7F. The authors state that PC1 is exclusively expressed in β -EP positive neurons. However, the only visible neurons are positive for both, so it would be helpful to see a nuclear or neuronal stain to see what percentage of neurons are positive for both (i.e. confirm that β -EP negative neurons don't have PC1).

Point-by-point response to reviewer comments:

Reviewer #1 (Remarks to the Author):

The manuscript by Tao and colleagues identify a novel epigenetic mechanisms underlying endogenous β -EP synthesis driven by microRNA regulation in ARC neurons involved in the pathogenesis of neuropathic pain. The manuscript is well-written and experiments are well designed. Results obtained in behavioral and molecular biology experiments are very consistent. The identification of epigenetic modifiers, such as microRNA-203a-3p, provides potential targets for development of chronic neuropathic pain.

However, there are some concerns as follows:

Author reply: We sincerely appreciate the referee's expertise and insightful comments, which are very helpful for revising and improving our manuscript.

One concern is that whether modulation of these epigenetic modifiers would lead to any potential adverse effects on the body. This aspect could be speculated in the discussion.

Author reply: We thank the reviewer for the helpful and constructive suggestion and have added the following paragraph in the *Discussion*.

“Nevertheless, the administration of epigenetic drugs necessitates caution, as it can be akin to a double-edged sword, potentially yielding unexpected side effects. It is imperative to acknowledge that the interventions of epigenetic modifiers such as NR4A2, H3K18ac, and miR-203a-3p may trigger the activation of genes associated with cognitive disorders (Engmann *et al.*, Brain, 2011) and proinflammatory genes (McEvoy *et al.*, Front Immunol, 2017; Ma *et al.*, Eur Rev Med Pharmacol Sci, 2020). Given the intricate and diverse nature of epigenetic modifications and their combinations, a careful assessment of epigenetic treatments is essential. The cellular specificity and safety and of miR-203a-3p and its modifiers will require further evaluation to address any potential off target activity.”

The relevant references have been cited and added to the reference list.

We note that only male rats were used in the manuscript, I recommend either being very transparent about this limitation in the manuscript or replicating one or two of key findings using female rats.

Author reply: We appreciate the reviewer's suggestion and additional experiments using female rats have now been carried out with these new data presented in the revised manuscript. As shown in Fig. 2, bilateral intra-ARC administration of miR-203a-down at Day 14 post-CCI-ION markedly alleviated mechanical allodynia as manifested by an increase in escape threshold to mechanical stimulation from Day 3 through Day 10 post drug administration in both male (Fig. 2B) and female rats (Fig. 2C). In addition, PCSK1-up microinjection attenuated nerve injury-induced mechanical allodynia from Day 3 through Day 10 after PCSK1-up administration in both male (Fig. 6F) and female rats (Fig. 6G). These results and descriptions have been added in *Materials and methods*, *Results* and *Figure legends*.

In addition, the present study demonstrated that blockade of miR-203a-3p by miR-203a-down significantly attenuated neuropathic pain in both male and female rats. Specific pathways in spinal microglia and sensory macrophages have been shown involved in the sexual dimorphism of neuropathic pain (Luo *et al.*, 2019; Midavaine *et al.*, 2021). Notably, sexual dimorphism may be limited to microglial and/or macrophages since the inhibition of pain-related signaling in neurons and astrocytes produced similar analgesia in both sexes (see Chen *et al.*, *Neurosci Bull*, 2018; Villa *et al.*, *Front Neuroendocrinol*, 2019). Given that miR-203a-3p, downstream PC1, as well as its cleavage product β -EP, are expressed exclusively in ARC neurons, it accounts for the consistency of male and female rat pain symptoms observed in the present study. Relevant descriptions concerning this issue have been included in the revised *Discussion*.

As a chronic neuropathic pain model, the CCI-ION model was used here. If the authors could replicate one or two key findings using 1-2 other neuropathic pain models, the evidence supporting these epigenetic modifiers as potential targets would be stronger.

Author reply: We agree and thank the reviewer for the constructive suggestion. As suggested, additional experiments have now been performed in the spinal nerve ligation (SNL) model. As shown in Fig. 2H, rats exhibited a significant decrease ($p < 0.001$) in paw withdrawal

threshold (PWT) from Day 3 through Day 21 after SNL compared to the counterparts following sham surgery. Bilateral intra-ARC injection of miR-203a-down at Day 7 post-SNL significantly attenuated mechanical allodynia from Day 3 through Day 10 post drug administration (Fig. 2I). Moreover, intra-ARC injection of PCSK1-up alleviated SNL-induced mechanical allodynia from Day 3 through Day 10 after PCSK1-up administration (Fig. 6I). These results and relevant descriptions have been added in *Materials and methods*, *Results*, and *Figure legends*.

It would be better to provide the qualitative analysis data showing statistic counting of the fluorescent expression neurons, for example, Fos-positive neurons in Figure 1B, as well as counting the percentage of miR-203a-3p-positive cells colocalized with neuronal markers such as in Figure 1K, 3H.

Author reply: As suggested, quantitative data analyses have been performed on the fluorescent images.

The expression of immediate early genes was analyzed from CCI-ION Day 14 to determine whether neuronal activity in ARC regions was altered. Notably, the number of neurons expressing Fos protein (Fos⁺) in the ARC was markedly elevated bilaterally following CCI-ION compared to the sham-operated groups (left, increased by $113.9 \pm 29.5\%$; right, increased by $121.2 \pm 22.3\%$; Fig. 1B). In addition, FISH analysis combined with immunofluorescent staining revealed that miR-203a-3p-positive cells primarily colocalized with the neuronal marker NeuN, but rarely with the astrocyte marker glial fibrillary acidic protein (GFAP) or the microglial marker integrin CD11b (Fig. 1L). Statistical analysis showed that ~93.4% of the miR-203a-3p⁺ neurons were labeled by NeuN, ~1.9% by GFAP, and ~3.7% by CD11b (Fig. 1L), indicating the predominant expression of miR-203a-3p in ARC neurons. Moreover, as shown in Fig. 3H, while miR-203a-3p expression was increased in the rat ARC on Day 14 following CCI-ION (increased by $78.4 \pm 17.3\%$), immunofluorescent staining revealed that the expression of NR4A2 appeared unchanged (increased by $-5.8 \pm 4.5\%$) (Fig. 3H). Statistical analysis showed that $23.8 \pm 2.6\%$ of DAPI-labeled ARC cells were positive for both miR-203a-3p and NR4A2 in sham surgery, while $41.7 \pm 4.3\%$ in CCI-ION rats (Fig. 3H).

Moreover, the qualitative analysis data showing statistic counting of the fluorescent expression neurons have now been provided in Figs. 4B and F, Fig. 6D and Fig. 7F, respectively. These results and relevant descriptions have been added in **Results** and **Figure legends**.

In line 110, the authors found that miR-203a-3p in the ARC was comparatively higher relative to other brain areas, but the ARC was not the highest area. As shown in Figure 1F, the highest area was NAc, and HIP was in comparable level with ARC. The authors could explain the potential effects of these areas.

Author reply: We thank the reviewer for the valuable suggestion (also raised by Reviewer #2). As a complimentary support of our hypothesis, an additional experiment was performed where we determined the expression levels of miR-203a-3p in the arcuate nucleus (ARC), nucleus accumbens (NAc), and hippocampus (HIP) of rats subjected to CCI-ION. In comparison to sham-operated groups, the expression levels of miR-203a-3p were markedly increased only in the ARC on Day 14 post-CCI-ION, but not in the NAc or HIP, although both areas possessed a high basal level of miR-203a-3p expression (Fig. 1G). These results and relevant descriptions have been added in **Results** and **Figure legends**.

In addition, as suggested, we have added the following paragraph in the **Discussion** to explain the potential effects of miR-203a-3p in the areas of NAc and HIP.

“Human imaging studies revealed that the NAc plays pivotal roles in both acute and chronic pain states (Baliki *et al.*, *Neuron*, 2010). Indeed, inactivating the NAc with lidocaine diminishes tactile allodynia in the spared nerve injury model of neuropathic pain (Chang *et al.*, *Pain*, 2014). In addition, manipulation of the HIP participates in both the processing and the modification of nociception (Bushnell *et al.*, *Nat Rev Neurosci* 2013; Wei *et al.*, *Pain*, 2021). However, it should be noted that β -EP-producing neurons are mainly clustered in the ARC (Sun *et al.*, *Pain*, 2003), while comparatively fewer in the HIP (Bansal *et al.*, *Front Cell Neurosci*, 2019) or NAc (Soderman and Unterwald, *Brain Res*, 2009). Importantly, although NAc and HIP had a high basal level of miR-203a-3p expression, comparative to ARC, the expression levels of miR-203a-3p remained unchanged in NAc and HIP regions after CCI-ION treatment (Fig. 1G). As a result, although both brain areas have potential involvement in pain regulation, the miR-203a-3p-mediated β -EP regulation seems less likely to function importantly in both

nuclei to regulate neuropathic pain. Nonetheless, further study is required to determine whether and how the two nuclei are involved in the regulation of trigeminal neuropathic pain.”

The relevant references have been cited and added to the reference list.

Another concern is that whether the antagomir or agomir used in this study was specific to manipulate miR-203a-3p. If not, it could not exclude the effect of other miRNAs. The authors used neuron-specific promoter lenti-hSyn-miR-203a-up to increase miR-203a expression. These are sufficient experiments to demonstrate the miR-203a-3p played a pivotal role in the neuropathic pain. It would be better if the authors could perform the necessary experiments with neuron-specific manipulation to decrease or block the expression of miR-203a in CCI-ION mice and test whether the manipulation would induce analgesia.

Author reply: We appreciate the reviewer for pointing out this concern. In this study, agomir, antagomir, or siRNA sequences were BLAST against the rat GenBank (<https://blast.ncbi.nlm.nih.gov/>) to exclude the presence of multiple target sequences in the rat genome. The sequences are provided in Table 3. The antagomir/agomir used in the study was specific to manipulate miR-203a-3p. As suggested, relevant descriptions have now been added in the *Materials and methods*. We further note that the antagomir/agomir sequences used in the present study are consistent with previous studies in which the antagomir/agomir has been shown to be specific to miR-203a-3p (Chang *et al.*, EMBO J, 2013; Li *et al.*, J Neuroinflammation, 2022).

Moreover, we overexpressed miR-203a-3p target sequences by employing a neuron promoter-specific combinatorial lentiviral vector lenti-hSyn-miR-203a-3p-antisense (miR-203a-down), which contained an eGFP construct serving as an expression marker, to investigate the effects of miR-203a-3p blockade on nociceptive behaviors (also raised by Reviewer #2). qPCR analysis revealed that no significant differences in the miR-203a-3p expression levels were observed in the ARC of miR-203a-down-treated CCI-ION rats (*Fig. S2*). Although the miRNA target sequence tightly binds to miR-203a-3p (Chang *et al.*, EMBO J, 2013), application of miR-203a-down did not induce the degradation of its complementary miRNA. Indeed, consistent with our present findings, it has been suggested that knockdown of miRNAs by overexpressing miRNA target sequences in lentiviral vectors does not affect the expression

levels of miRNAs (Gentner *et al.*, Nature Methods, 2009; Bak *et al.*, Molecular Therapy, 2013). Moreover, bilateral intra-ARC administration of miR-203a-down at Day 14 post-CCI-ION markedly alleviated mechanical allodynia from Day 3 through Day 10 post drug administration in both male (Fig. 2B) and female rats (Fig. 2C). As suggested, the previously somewhat obscure and confusing statements in *Materials and methods*, *Results* and *Figure legends* have now been corrected.

Minor details:

Grammar mistakes, for example in line 109

Author reply: We appreciate the reviewer for pointing out this concern. As suggested, the poor grammar issues have been corrected.

Figure 1B: “ham” should be “sham”

Author reply: The typo previously in Fig. 1B has now been corrected.

Figure 1C: less of description of the color bar

Author reply: We thank the reviewer for pointing this out. In Fig. 1C, color bar (*up left*) indicates the scale of standardized miRNA levels. Warm color indicates higher concentration. The relevant descriptions have now been added in the legend of Fig. 1C.

Figure 1D: inconsistent in star character size

Author reply: The character size of the stars in Fig. 1D are now presented consistently.

Figure 4K: the quality of the blots was not good. It would be better if the authors could provide better images.

Author reply: We appreciate the reviewer for pointing out this concern. As suggested, higher quality blots are provided in Fig. 4K. In addition, the full-length blots have been provided in Supplementary Fig. S13.

Reviewer #2 (Remarks to the Author):

Central to this manuscript is the elucidation of the role of miR-203a-3p that is overexpressed in the hypothalamic ARC neurons, much lower in astrocytes or microglia after chronic constriction ligation of the infraorbital nerve to model neuropathic pain in the rat. The manuscript then described the transcriptional regulation of miR-203a-3p expression by transcriptional factor NR4A2 via epigenetic regulation of acetylation of the miR-203a-3p promoter by HDAC9, while NR4A2 protein level was similar between sham and CCI-ION rats. After, determining the upstream regulation of miR-203a-3p transcript expression, the authors examined the downstream target of the ncRNA and found that the PCSK1 gene that codes for PC1 was downregulated by it. The authors then determined the potential link between PC1 downregulation and b-EP production from POMC in the ARC neurons and related their work to mechanical allodynia.

Overall, the experiments were well designed and the rationale for selecting which transcriptional factor or acetylase or deacetylase to focus on was reasonable and tested in well controlled experiments.

Could the authors address the following comments and suggestions with the aim to improve the clarity and depth of the discussion:

Author reply: We sincerely appreciate the referee's expertise and insightful comments, which are very helpful for revising and improving our manuscript.

1. High levels of miRNA 203a-3p were also detected in brain regions like hippocampus and NAc (Fig 1F). As NAc also plays an important role in pain processing, is the current mechanism also applicable to NAc? Was miRNA 203a-3p also increased in NAc post CCI and could the authors comment on this in Discussion?

Author reply: We thank the reviewer for the valuable suggestion. As a complimentary support of our hypothesis, an additional experiment was performed where we determined the expression levels of miR-203a-3p in the arcuate nucleus (ARC), nucleus accumbens (NAc), and hippocampus (HIP) of rats subjected to CCI-ION. In comparison to sham-operated groups, the expression levels of miR-203a-3p were markedly increased only in the ARC on Day 14 post-CCI-ION, but not in the NAc or HIP, although both areas possessed a high basal level of miR-

203a-3p expression (Fig. 1G). These results and relevant descriptions have been added in **Results** and **Figure legends**.

In addition, as suggested (also raised by Reviewer #1), we have added the following paragraph in the **Discussion** to explain the potential effects of miR-203a-3p in the areas of NAc and HIP.

“Human imaging studies revealed that the NAc plays pivotal roles in both acute and chronic pain states (Baliki *et al.*, Neuron, 2010). Indeed, inactivating the NAc with lidocaine diminishes tactile allodynia in the spared nerve injury model of neuropathic pain (Chang *et al.*, Pain, 2014). In addition, manipulation of the HIP participates in both the processing and the modification of nociception (Bushnell *et al.*, Nat Rev Neurosci 2013; Wei *et al.*, Pain, 2021). However, it should be noted that β -EP-producing neurons are mainly clustered in the ARC (Sun *et al.*, Pain, 2003), while comparatively fewer in the HIP (Bansal *et al.*, Front Cell Neurosci, 2019) or NAc (Soderman and Unterwald, Brain Res, 2009). Importantly, although NAc and HIP had a high basal level of miR-203a-3p expression, comparative to ARC, the expression levels of miR-203a-3p remained unchanged in NAc and HIP regions after CCI-ION treatment (Fig. 1G). As a result, although both brain areas have potential involvement in pain regulation, the miR-203a-3p-mediated β -EP regulation seems less likely to function importantly in both nuclei to regulate neuropathic pain. Nonetheless, further study is required to determine whether and how the two nuclei are involved in the regulation of trigeminal neuropathic pain.”

The relevant references have been cited and added to the reference list.

2. POMC is known to be circadian rhythmically regulated. As such, what ZT time were all the samples collected for detection of POMC and PC1 levels?

Author reply: We agree with the comment and appreciate the reviewer in pointing out this issue. Zeitgeber time (ZT) is a measure of time (hours) after lights go on, in which ZT0 being the “lights on” onset and ZT12 being the onset of “lights off”. In our study, all the tissues/samples were collected between ZT6 and ZT8 for detection of POMC and PC1 levels, since the expression level of POMC is reported to be relatively stable during this time period (Agapito *et al.*, Endocrinology, 2014). In addition, to further avoid potential circadian rhythmically regulated effects on POMC or PC1 levels, we strictly set the corresponding control groups in

which tissues/samples were collected using the same time period of ZT6 to ZT8. These descriptions have now been added in the *Materials and methods*.

3. Could the authors comment on the increased level of POMC in the CCI-ION rats. What role does an increase in ARC POMC serve as later its cleavage will be decreased via an epigenetic mechanism that downregulates PC1, and the eventual pathway will subsequently lead to less β -EP being generated? Is this a compensatory or neutral process?

Author reply: We thank the reviewer for pointing out this issue and we have performed an additional experiment in these regards. qPCR analysis showed that the mRNA level of POMC remained unchanged in rat ARC on Day 14 after CCI-ION (*Fig. S23*), indicating that nerve injury did not induce any changes in POMC synthesis. However, our results show that POMC protein expression is significantly increased in the rat ARC from 7 days to 28 days post-CCI-ION, but remains unaffected in the sham-operated groups (*Figs. 7B and C & Fig. S24*). In the ARC of CCI-ION rats, since POMC synthesis (mRNA level) remained unchanged, a decrease in PC1 led to less cleavage of POMC, consequently causing an accumulation of POMC. We have included these results and descriptions in *Results, Discussion* and *Figure legends*.

4. β -EP is also known to be involved in stress. A recent study has also shown that chronic restrained stress decreased the excitability of POMC neurons. Could the increase of miRNA 203a-3p also be due to the stress response induced by CCI?

Author reply: We appreciate the reviewer for the helpful suggestion and have added the following paragraph to the *Discussion*.

“ β -EP is also known to be involved in stress (Marinelli *et al.*, Neuroscience, 2004). For instance, physical and fearing-inducing psychological stressors stimulate β -EP release in the ARC (Marinelli *et al.*, Neuroscience, 2004). In addition, a recent study showed that chronic restraint stress decreases the excitability of hypothalamic POMC neurons (Ha and Cheong, *Exp Neurobiol*, 2021), together suggesting a potential role of stress in β -EP regulation. Here, we found that nerve injury upregulated miR-203a-3p in the ARC, resulting in a decrease of β -EP. Interestingly, it has been shown that chronic stress did not induce changes of miR-203a-3p expression in the HIP and medial prefrontal cortex (Babenko *et al.*, Plos One, 2011; An *et al.*,

Biochem Biophys Res Commun, 2020). Additionally, injection of chemical stressors did not elicit changes of β -EP expression in the ARC (Marinelli *et al.*, Neuroscience, 2004). Although stressors with different properties are processed differently in the brain (Marinelli *et al.*, Neuroscience, 2004), it can be speculated that increased miR-203a-3p in the ARC was less likely due to the stress response induced by CCI-ION. Consistent with this notion, previous studies have revealed that chronic stress had no influence on mechanical threshold (Bravo *et al.*, Anesthesiology, 2012; Le Coz *et al.*, Front Behav Neurosci. 2017). Nevertheless, chronic pain and stress are two complex states, and they may mutually exacerbate one another in conditions of comorbidity. Further investigation is necessary to examine the role of stress in trigeminal-mediated neuropathic pain; however, we believe it is beyond the scope of the current study.”

The relevant references have been cited and added to the reference list.

5. In all the experiments with injections of siRNA or agomir/antagomir-203a to ARC, how did the authors ensure the effect is specific to ARC and that any spreading effect did not affect the neighbouring brain regions?

Author reply: We appreciate the reviewer in pointing out this concern. Stereotaxic intracerebral cannula implantation was conducted for drug/reagent administration with ARC coordinates determined by reference to the rat brain in stereotaxic coordinates as per George Paxinos & Charles Watson. Rats were injected using a twenty-gauge microinjection needle (Hamilton) with 0.5 μ l of reagents. The injection sites within the ARC were histologically verified afterward by injecting blue dye which was mixed with the injection solutions so that accurate targeting of the ARC could be monitored. As shown in *Fig. S25*, blue color appeared within the ARC area with no apparent spreading of the injected reagents into the surrounding areas, ensuring the effects of reagents being specific to ARC without any spreading effect on the neighboring brain regions.

Moreover, we examined GFP expression in rats intra-ARC injected with lenti-hSyn-miR-203a-3p-up containing an eGFP construct that served as an expression marker (also raised by Reviewer #3). As shown in Fig. 2F, almost all GFP-expressing neurons with green fluorescence were located within the ARC (outlined by white dashed line), while the surrounding areas

showed little expression, further supporting the effects of drugs/reagents specific to ARC. As suggested, the zoomed out image of the entire ARC area has now been shown in Fig. 2F. These results and descriptions have been added in *Materials and methods*, *Results* and *Figure legends*.

6. The authors showed that down-regulation of miR-203a-3p would reduce nociceptive behaviours in the CCI-ION rats. If using a pharmacological method to relief pain, would the level of miR203a-3p change in response to the drug?

Author reply: We appreciate the reviewer in pointing this out. In this study, we overexpressed miR-203a-3p target sequences by employing a neuron promoter-specific combinatorial lentiviral vector lenti-hSyn-miR-203a-3p-antisense (miR-203a-down) carrying eGFP gene to investigate the effects of miR-203a-3p blockade on nociceptive behaviors (also raised by Reviewer 1). qPCR analysis revealed that no significant differences in the miR-203a-3p expression levels were observed in the ARC of miR-203a-down-treated CCI-ION rats (*Fig. S2*). Although the miRNA target sequence tightly binds to miR-203a-3p (Chang *et al.*, EMBO J, 2013), application of miR-203a-down did not induce the degradation of its complementary miRNA. Indeed, consistent with our findings, it has been suggested that knockdown of miRNAs by overexpressing miRNA target sequences in lentiviral vectors does not affect the expression levels of miRNAs (Gentner *et al.*, Nature Methods, 2009; Bak *et al.*, Molecular Therapy, 2013). Moreover, bilateral intra-ARC administration of miR-203a-down at Day 14 post-CCI-ION markedly alleviated mechanical allodynia from Day 3 through Day 10 post drug administration in both male (*Fig. 2B*) and female rats (*Fig. 2C*). These results and descriptions have been included in *Materials and methods*, *Results* and *Figure legends*.

7. When using POMC-siRNA, multiple injections were performed (*Fig 7M*). What is the reason for injecting three times? For the same *Fig 7M*, is the injection of miR203a-down applied to all groups or the CCI-ION rats only? Please clarify in the legend.

Author reply: We appreciate the reviewer in pointing out this concern. Since β -EP is the cleavage product of POMC, we applied siRNA knockdown of POMC to abolish β -EP synthesis. We intra-TG injected 5'-Cholesteryl-modified and 2'-O-methyl-modified siRNA (RiboBio

Biological Technology). This chemical modification can enhance cell membrane penetration (5'-Cholesteryl-modified) and prevent degradation (2'-O-methyl-modified) of siRNAs. Nonetheless, behavioral data (as indicated in Fig. 4N and Fig. 6K) showed that chemically modified siRNA injection generally had the best knockdown effect on the 3rd or 4th day, and gradually recovered after that time. In order to avoid the pain threshold recovery caused by the partial degradation of siRNA, we performed multiple injections of POMC-siRNA (intra-ARC on Days 0, 3 and 6) to ensure that the maximum inhibition of POMC expression was maintained after miR-203a-down application (Day 3 to Day 10 in Fig. 7M). Further intra-ARC injection of miR-203a-down (Day 3) markedly attenuated mechanical allodynia in CCI-ION rats pretreated with NC-siRNA, while had no effects in CCI-ION rats subjected to POMC-siRNA (Fig. 7M). The injection of miR-203a-down was applied to all the four groups in Fig. 7M. As suggested, relevant descriptions have been included in the legend of Fig. 7M.

Minor Comments:

1. Title: would authors consider using “modulates” instead of “controls”, as epigenetic phenomenon is dynamic.

Author reply: We agree with the reviewer and as suggested, the title has been revised to “Epigenetic regulation of beta-endorphin synthesis in hypothalamic arcuate nucleus neurons modulates neuropathic pain”.

2. Line 35: suggest changing “...increased acetylation” to “decreased deacetylation”, as this is a more intuitive description.

Author reply: As suggested, “increased acetylation” has now been changed to “decreased deacetylation” in the **Abstract**.

3. Line 37: what does “upstream” mean in the sentence. Please be clear.

Author reply: As suggested, the previously somewhat obscure statement has been revised for clarity.

4. Line 37: The sentence beginning with “The increased miR-203a-3p.....” is structured awkwardly and I suggest the authors rephrase.

Author reply: The mentioned sentence has now been rephrased as following: “Further, increased miR-203a-3p was found to maintain neuropathic pain by targeting proprotein convertase 1, an endopeptidase necessary for the cleavage of proopiomelanocortin, the precursor of β -EP.”

5. Lines 40 and 466: change “revenue” to “an avenue”

Author reply: Changed as suggested.

6. Line 46: should it not be ... lesions and diseases of somatosensory nerves rather than nervous system?

Author reply: We thank the reviewer for pointing out this concern. As suggested, the relevant description has been addressed.

7. Line 50: suggest to use the word “modulation” instead of “pathogenesis”.

Author reply: As suggested, the word “modulation” has now been used in the text.

8. Line 80: alteration instead of alternation

Author reply: The typo has now been corrected.

Reviewer #3 (Remarks to the Author):

Tao et al identify a new epigenetic pathway regulating beta-endorphin regulation in neuropathic pain. The authors do a good job of systematically following the pathway from regulatory microRNAs all the way to beta-endorphin regulation in a model of neuropathic pain. They use a series of straight-forward approaches to show that miR-203a-3p is upregulated in a model of chronic pain and in human patients, and that this miR promotes nociceptive behavior by regulating levels PCSK1. In addition, they show that miR-203a-3p expression is regulated by

the transcription factor NR4A2, whose binding is regulated by HDAC9. Overall, I think this is a comprehensive series of experiments that elucidates a new pathway of pain regulation. I have several minor comments.

Author reply: We sincerely appreciate the referee's expertise and insightful comments, which are very helpful for revising and improving our manuscript.

Minor comments

Figure 2E. Please add zoomed out image of ARC to see extent/spread of lentiviral infection.

Author reply: We thank the reviewer in pointing out this concern. We examined GFP expression in rats intra-ARC injected with lenti-hSyn-miR-203a-3p-up containing an eGFP construct serving as an expression marker (also raised by Reviewer #2). As shown in Fig. 2F, almost all eGFP-expressing neurons with green fluorescence were located within the ARC (outlined by white dashed line), while the surrounding areas showed little expression, supporting the effects of drugs/reagents being specific to ARC. As suggested, the zoomed out image of the entire ARC area is now shown in Fig. 2F. The results and descriptions have been added in *Materials and methods*, *Results*, and *Figure legends*.

Figure 3H. How many rats were used for IHC/in situ?

Author reply: Four rats were used in each group. This information has now been added in the legend of Fig. 3H.

Figure 4A. what were the western blots normalized to? H3 mods are all about the same weight as total H3 and since blots were labeled with chemiluminescence, it's not clear how they could both be run on the same membrane to allow for normalization. Were the blots stripped? If so, what was the stripping protocol and in what order were the antibodies incubated? In other figures, GAPDH is used to normalize, but staining usually shows just the GAPDH band, so it's also not clear if these were done on the same membrane?

Author reply: We thank the reviewer for pointing out this concern. In the present study, histone mods were normalized to total H3, and other proteins were normalized to GAPDH. Histone mods were separated on SDS-PAGE gels. After exposed to primary monoclonal antibodies,

bound proteins were incubated with relevant second antibodies. Subsequently, the membranes blotted for acetylation-modified histone were stripped with Western Blot Stripping Buffer (Beyotime) for 10 min, washed three times with TBST, blocked again with 5% nonfat milk, and re-probed with primary antibodies against total H3 proteins, an internal control for ac-H3. As suggested, these results and descriptions are included in *Materials and methods*. In addition, the target proteins and GAPDH were performed on the same membrane without stitching. The full-length blots are now provided in Supplementary Fig. S8. All unedited blots (with molecular weights) and gel images have been included in the *Supporting Information*.

Figure 4C. Please clarify how ChIP quantification was normalized. Figure captions say that data are normalized to input, but it looks like they may also be normalizing to sham?

Author reply: We thank the reviewer for pointing out this issue. The ChIP assays were quantified by qPCR. The fold enrichment of ChIP data was normalized to the input DNA and then compared to the corresponding control groups (sham in Fig. 3I & Fig. 4C, CCI-ION+HDAC9-NC in Fig. 4I, or NC-siRNA in Fig. 4M). These descriptions have included in the legend of Fig. 3I, Fig. 4C, Fig. 4I, and Fig. 4M.

Figure 6C. The authors report a correlation between PCSK1 expression and miR-203a expression. Why are there 2 clusters? If this reflects a group difference between CCI-ION vs control, data should be shown as bar graph, not correlation. Figure caption states that this figure shows protein abundance at day 0, 7, 14, 21, and 28, so the correlation may reflect differences between miR-203a-3p and PC1 levels over this time period? This is not clear in the text. Either way, use of correlations to describe 2 distinct data clusters seems inappropriate

Author reply: We thank the reviewer for pointing this out. Both of the bar graphs revealed the group differences of miR-203a (Fig. 1J) and PC1 protein expression (Fig. 6A) between CCI-ION and sham-operated rats, respectively. We agree with the reviewer that applying two different data clusters for correlation analysis is not suitable. Thus, as shown in Fig. 6C, the decreased mRNA level of PCSK1 at day 0, 7, 14, 21, and 28 (Fig. S18) showed an inverse correlation with miR-203a-3p upregulation in the ARC of CCI-ION rats. The previous

inaccurate statements were corrected and results and descriptions have been added in **Results** and **Figure legends**.

Figure 7F. The authors state that PC1 is exclusively expressed in β -EP positive neurons. However, the only visible neurons are positive for both, so it would be helpful to see a nuclear or neuronal stain to see what percentage of neurons are positive for both (i.e. confirm that β -EP negative neurons don't have PC1).

Author reply: We agree and thank the reviewer for pointing out this concern. As suggested, additional experiments as well as data analyses have been performed to address the issue raised. As shown in Fig. 7F, immunostaining indicated that PC1 was almost exclusively expressed in β -EP-positive ARC neurons in sham- and CCI-ION-operated rats (Fig. 7F). Notably, β -EP-negative ARC cells did not express PC1 (stained by nuclear marker DAPI). Statistical analysis showed that approximately $25.6 \pm 1.9\%$ of DAPI-labeled ARC cells were positive for both β -EP and PC1 in sham-operated rats, while $14.1 \pm 1.5\%$ in CCI-ION rats (Fig. 7F). These results and descriptions are now included in the **Results** and **Figure legends**, respectively.

In summary, the manuscript has been thoroughly revised and amended, including new data from the experiments suggested by the editor and reviewers. The relevant references are also cited and included in the reference list.

REVIEWERS' COMMENTS

Reviewer #1 (Remarks to the Author):

The study is, in its revised form, important and convincing.
My major and minor concerns were well answered.

This manuscript is recommended for acceptance.

Reviewer #2 (Remarks to the Author):

The authors have satisfactorily addressed my concerns.

Reviewer #3 (Remarks to the Author):

The authors conducted an extensive revision to address reviewer comments. I don't have any additional concerns.

Point-by-point response to the reviewers' comments:

REVIEWERS' COMMENTS

Reviewer #1 (Remarks to the Author):

The study is, in its revised form, important and convincing. My major and minor concerns were well answered. This manuscript is recommended for acceptance.

Author reply: We are very grateful for the reviewer's positive comments.

Reviewer #2 (Remarks to the Author):

The authors have satisfactorily addressed my concerns.

Author reply: We are very grateful for the reviewer's positive comments.

Reviewer #3 (Remarks to the Author):

The authors conducted an extensive revision to address reviewer comments. I don't have any additional concerns.

Author reply: We are very grateful for the reviewer's positive comments.